

# A Moist Quasi-Geostrophic Coupled Model: MQ-GCM2.0

Sergey Kravtsov[1,2,3], Ilijana Mastilovic[1], Andrew McC. Hogg[4], William K. Dewar[5,6] and Jeffrey R. Blundell[7]

[1]Department of Mathematical Sciences, University of Wisconsin-Milwaukee, P. O. Box 413, Milwaukee, WI 53201, USA
[2]Shirshov Institute of Oceanology, Russian Academy of Sciences, Moscow, 117218, Russia
[3]Institute of Applied Physics, Russian Academy of Sciences, Nizhniy Novgorod, 603155, Russia
[4]Research School of Earth Sciences, and ARC Centre of Excellence in Climate Extremes, Australian National University, Canberra, Australia.
[5]Department of Earth, Ocean and Atmospheric Science, Florida State University, Tallahassee, FL 32304, USA
[6]Laboratoire de Glaciologie et Geophysique de l'Environnement, CNRS, Grenoble, France
[7]Ocean and Earth Science, National Oceanography Centre Southampton, University of Southampton, Southampton SO14 3ZH, United Kingdom

*Correspondence to*: Sergey Kravtsov (kravtsov@uwm.edu)

**Abstract.** This paper contains a description of recent changes to the formulation and numerical implementation of the Quasi-Geostrophic Coupled Model (Q-GCM), which constitute a major update of the previous version of the model (Hogg et al., 2014). The Q-GCM model has been designed to provide an efficient numerical tool to study the dynamics of multi-scale mid-latitude air–sea interactions and their climatic impacts. The present additions/alterations were motivated by an inquiry into the dynamics of mesoscale ocean–atmosphere coupling and, in particular, by an apparent lack of Q-GCM atmosphere's sensitivity to mesoscale sea-surface temperature (SST) anomalies, even at high (mesoscale) atmospheric resolutions, contrary to ample theoretical and observational evidence otherwise. Major modifications aimed at alleviating this problem include an improved radiative-convective scheme resulting in a more realistic model mean state and associated model parameters, a new formulation of entrainment in the atmosphere, which prompts more efficient communication between the atmospheric mixed layer and free troposphere, as well as an addition of temperature-dependent wind component in the atmospheric mixed layer and the resulting mesoscale feedbacks. The most drastic change is, however, the inclusion of moist dynamics in the model, which may be key to midlatitude ocean–atmosphere coupling. Accordingly, this version of the model is to be referred to as the MQ-GCM model. Overall, the MQ-GCM model is shown to exhibit a rich spectrum of behaviours reminiscent of many of the observed properties of the Earth's climate system. It remains to be seen whether the added processes are able to affect in fundamental ways the simulated dynamics of the mid-latitude ocean–atmosphere system's coupled decadal variability.

## 1 Introduction

The Quasi-Geostrophic Coupled Model (Q-GCM) was initially developed by Hogg et al. (2003) and has been substantially modified since: its latest distribution and source code are publicly available at http://www.q-gcm.org and are fully


documented in the Q-GCM Users' Guide, v1.5.0 (Hogg et al., 2014). The model couples the multi-layer quasi-geostrophic (QG) ocean and atmosphere components via ageostrophic mixed layers that regulate the exchange of heat and momentum between the two fluids. Q-GCM model can be configured as either a box (a 'double-gyre') or a channel ocean (a 'Southern Ocean') underneath a channel atmosphere; it conceptualizes the mid-latitude climate system driven by the latitudinal variation of the incoming solar radiation. In addition to the oceanic mixed layer, the model physics incorporates a

dynamically active atmospheric mixed layer (effectively, the atmospheric planetary boundary layer: APBL), the dependence of the wind stress on the ocean–atmosphere surface velocity difference, as well as a dynamically consistent parameterization of the entrainment heat fluxes between the model layers. It can also be easily modified to include a parameterization of a sea-surface temperature (SST) feedback on the wind stress (e.g., Hogg et al., 2009), which will be a part of the new version of the model developed here. Q-GCM thus encompasses a richer, more comprehensive set of processes, enabling one to achieve

a more accurate simulation of the ocean–atmosphere coupling, especially at mesoscales, relative to some other, analogous conceptual models, which either assume the atmospheric near-surface temperature to be in equilibrium with SST (e.g., Feliks et al., 2004, 2007, 2011; cf. Schneider and Qiu, 2015) or relate this temperature in an ad hoc way to the instantaneous in-situ distribution of the model's tropospheric temperature (Kravtsov et al. 2005, 2006, 2007; Deremble et al. 2012). The Q-GCM model was previously used for ocean-only and coupled experiments around the double-gyre problem (Hogg et al., 2005,

2006, 2007; Martin et al. 2020), as well as in the ocean-only studies of the Southern Ocean's climate system (Hogg and Blundell, 2006; Meredith and Hogg, 2006; Hogg et al., 2008; Hutchinson et al., 2010; Kravtsov et al., 2011).

The long oceanic thermal and dynamical inertia makes the ocean a primary agent for generating potentially predictable climate signals on time scales from years to decades, whereas atmospheric intrinsic time scales are significantly shorter. The null hypothesis for climate variability views the ocean as a passive integrator of high-frequency noise associated with

atmospheric geostrophic turbulence (Hasselmann, 1976; Frankignoul and Hasselmann, 1977; Frankignoul, 1985; Barsugli and Battisti, 1998; Xie, 2004). However, both observations (e.g., Chelton, 2013; Frenger et al., 2013) and decades of experimentation with wind-driven eddy-resolving ocean models (e.g., Berloff and McWilliams, 1999; Primeau, 2002; Berloff et al., 2007; Hogg et al., 2009; Shevchenko et al., 2016, among many others) documented vigorous internal variability and the associated mesoscale features (fronts and eddies with spatial scales of 10–100km) throughout the World

Ocean. Two key regions in which these eddies are most important are near the western boundary currents and their extensions and in the Southern Ocean. Mesoscale variability in these regions modulates atmospheric fronts and storms' intensity and distribution, thus affecting atmospheric variability on short time scales (e.g., Maloney and Chelton, 2006; Minobe et al., 2008; Nakamura and Yamane, 2009; Bryan et al., 2010; Chelton and Xie, 2010; Kuwano-Yoshida et al., 2010; O'Neill et al., 2010, 2012; Frenger et al., 2013; O'Reilly and Czaja, 2015; Seo et al., 2016; Parfitt et al., 2017). Recent

observational and modelling evidence strongly suggested that this mesoscale oceanic turbulence may also imprint itself onto large-scale low-frequency climate modes (with time scales from intra-seasonal to decadal), which would have profound consequences for near-term climate predictability (e.g., Hogg et al., 2006; Siqueira and Kirtman, 2016). To study this





phenomenon may, therefore, require coupled climate models with high horizontal resolution in both their oceanic and atmospheric components (see below); this would make the requisite long climate simulations using highly resolved state-of-
the-art climate models computationally infeasible.

Feliks et al. (2004, 2007, 2011) and Brachet et al. (2012) examined the response of the atmosphere to mesoscale sea-surface temperature (SST) anomalies through hydrostatic pressure adjustment in an idealized atmospheric model. They showed that resolving an ocean front and mesoscale eddies affects atmospheric climatology, intraseasonal modes, as well as decadal variability (when forced with the observed SST history) in their model (see also Nakamura et al., 2008). These authors
argued that atmospheric components of global climate models must resolve oceanic fronts to faithfully simulate the observed climate variability (see also Minobe et al., 2008). Ma et al. (2017) also concluded that "It is only when the (atmospheric) model has sufficient resolution to resolve small-scale diabatic heating that the full effect of mesoscale SST forcing on the storm track can be correctly simulated," with the ensuing consequences for atmospheric low-frequency variability associated with the downstream Rossby wave breaking (Piazza et al., 2016) and blocking (O' Reilly et al., 2015). By contrast, Bryan et
al. (2010) proposed that accurate representation of mesoscale ocean–atmosphere coupling in a model depends more on the Marine Atmospheric Boundary Layer (MABL) mixing scheme than on the ability of an atmospheric model to resolve a thermal front per se; this would justify the use of atmospheric resolutions on the order of 50 km in many GCM studies that documented a pronounced influence of the mesoscale air–sea interactions on the atmospheric storm tracks (Miller and Schneider, 2000; Nakamura et al., 2008; Taguchi et al., 2009; Kelly et al., 2010; Perlin et al., 2014; Small et al., 2014; Ma et
al., 2015, 2017; Piazza et al., 2016).

A numerically efficient intermediate-complexity Q-GCM model thus provides an alternative (to highly resolved GCMs) and unique tool ideally suited to help advance our understanding of multi-scale ocean–atmosphere interactions and their climatic impacts. Its QG dynamical core resolves well the geostrophic turbulence on either side of the ocean–atmosphere interface, including oceanic mesoscale eddies/fronts and atmospheric storm tracks. The existing version of the coupled model,
however, lacks the parameterization of SST effects on the model's MABL winds. One such parameterization was tested in the Q-GCM's ocean-only configuration by Hogg et al. (2009). Mastilovic and Kravtsov (2019) examined the effects of both Hogg et al. (2009) and Feliks et al.'s (2004, 2007) SST-dependent MABL wind formulations in the context of coupled Q-GCM simulations with the standard (coarse) and fine (mesoscale resolving) atmospheric grid spacing. They found that — consistent with previous studies (Dewar and Flierl, 1987; Maloney and Chelton, 2006; Hogg et al. 2009; Gaube et al., 2013,
2015; Chelton, 2013; Small et al., 2014) — these effects constitute a negative feedback on the ocean and tend to reduce the intensity of the oceanic mesoscale perturbations that generated the mesoscale wind anomalies in the first place. In a coupled setting, this leads to a dilution of oceanic mesoscale features and the resulting lack of model's sensitivity to atmospheric resolution. Surprisingly, the *atmosphere-only* Q-GCM simulations with and without an SST front, or with and without SST-dependent AMBL winds, apparently also produce statistically identical atmospheric variability irrespective of the



atmospheric resolution (Mastilovic and Kravtsov 2020, personal communication), in sharp contrast to the resolution-dependent dynamics documented in Feliks et al. (2004, 2007). Furthermore, the entire previous experience with Q-GCM indicates the absence, in the existing version of the model, of a nonlinear, "weather-regime"-type atmospheric behaviour documented in analogous atmospheric (Marshall and Molteni, 1993; Kravtsov et al., 2005) and coupled models (Kravtsov et al., 2006, 2007); such behaviour may lead to a nonlinear atmospheric sensitivity to ocean induced SST anomalies and

generate fundamentally coupled decadal climate modes (e.g., Kravtsov et al., 2008).

The purpose of this paper is to document a new, revamped version of the Q-GCM model, which addresses an apparent lack of the Q-GCM atmosphere's sensitivity to SST anomalies, contrary to ample theoretical and observational evidence otherwise. In section 2, we briefly summarize the dynamical (QG) core of the model, placing some of the supporting information in the appendix. Major modifications to the original, dry-model physics (section 3) include an improved

radiative-convective scheme resulting in a more realistic model mean state and the associated model parameters, a new formulation of entrainment in the atmosphere, which prompts a more efficient communication between the atmospheric mixed layer and free troposphere, as well as an addition of temperature-dependent wind component in the atmospheric mixed layer and the resulting mesoscale feedbacks. The most drastic change is, however, the inclusion, in the model, of moist dynamics (section 4), which may be key to midlatitude ocean–atmosphere coupling (Czaja and Blunt, 2011; Laîné et

al., 2011; Deremble et al., 2012; Willison et al., 2013; Foussard et al., 2019). Accordingly, this version of the model is to be referred to as the MQ-GCM model. Overall, the MQ-GCM model is shown to exhibit a rich spectrum of behaviours reminiscent of many of the observed properties of the Earth's climate system (section 5). The paper concludes with some discussion in section 6 and section 7, which summarizes the MQ-GCM changes and code modifications with respect to the original Q-GCM version. This presentation is also supplemented with the collection of Fortran 90 routines containing all the

new MQ-GCM source code that complement the original Q-GCM distribution (http://www.q-gcm.org).

## 2 Q-GCM dynamical core

Q-GCM model incorporates quasi-geostrophic dynamics on a $\beta$-plane in its $n$-layer oceanic and atmospheric modules (below, we will use $n$=3); these dynamics are governed by the equations describing the evolution of quasi-geostrophic

potential vorticity $q$. For a flat-bottom ocean (used here for simplicity, although the topography is included in Q-GCM)

$$^{o}q_{k\,t} + J(^{o}\psi_k, {}^{o}q_k) = \frac{f_0}{{}^{o}H_k}({}^{o}e_{k-1} - {}^{o}e_k) + {}^{o}A_2 \nabla_H^4 {}^{o}\psi_k - {}^{o}A_4 \nabla_H^6 {}^{o}\psi_k, \ k = 1, 3, \tag{1}$$

where





$$^{o}q_{k} = \nabla_{H}^{2}\,{}^{o}\psi_{k} + \frac{f_{0}}{{}^{o}H_{k}}({}^{o}\eta_{k} - {}^{o}\eta_{k-1}) + \beta(y - y_{0}); \; k = 1,3 \tag{2}$$

and


$$^{o}\eta_{k} = \frac{f_{0}}{{}^{o}g'_{k}}({}^{o}\psi_{k+1} - {}^{o}\psi_{k}),\; {}^{o}g'_{k} = g\frac{({}^{o}\rho_{k+1} - {}^{o}\rho_{k})}{{}^{o}\rho_{0}}; \; k = 1,2; {}^{o}\eta_{0} = {}^{o}\eta_{3} = 0. \tag{3}$$

In the equations above, the left superscript 'o' refers to the oceanic quantities, $^{o}q_{k}$ is the potential vorticity for ocean layer $k$, counted from the surface down, $^{o}\psi_{k}$ is the layer-$k$ geostrophic streamfunction, $^{o}H_{k}$ are layer thicknesses and $^{o}\rho_{k}$ — layer densities (all close to the representative water density $^{o}\rho_{0}$ ), with $k = 1,3$. Furthermore, $^{o}\eta_{k}$ are the perturbation displacements of the interface between the top/middle and middle/bottom layers, $^{o}g'_{k}$ are reduced gravity coefficients (for

both of these $k = 1,2$ ), $f_{0}$ and $\beta$ are the Coriolis parameter and its $y$-derivative at the central latitude $y_{0}$, respectively, $^{o}A_{2}$ and $^{o}A_{4}$ are viscosity coefficients for the Laplacian and biharmonic friction parameterizations, the subscript $t$ denotes the time derivative, $\nabla_{H}^{2}$ is the horizontal Laplacian operator, and $J$ is the Jacobian operator. The ocean model is driven by the entrainment $^{o}e_{0} = {}^{o}w_{ek}$ associated with the surface Ekman pumping $^{o}w_{ek}$ computed as the curl of the wind stress; the model also includes Ekman dissipation at the bottom, with $^{o}e_{3} \sim \nabla_{H}^{2}\psi_{3}$, as well as a thermally driven entrainment $^{o}e_{1}$ (between

ocean layers 1 and 2) due to heat exchange between the ocean's layer-1 and the mixed layer (see section 3.2 below, as well as Hogg et al., 2014 for further details).

The atmospheric module mirrors the ocean module: it is set up in a fluid comprised of layers with constant potential temperatures $\theta_{k}$ and variable depths (see the appendix). In particular,


$$^{a}q_{k\,t} + J({}^{a}\psi_{k}, {}^{a}q_{k}) = \frac{f_{0}}{{}^{a}H_{k}}({}^{a}e_{k} - {}^{a}e_{k-1}) - {}^{o}A_{4}\nabla_{H}^{6}\,{}^{a}\psi_{k}, \; k = 1,3; \tag{4}$$

$$^{a}q_{k} = \nabla_{H}^{2}\,{}^{a}\psi_{k} + \frac{f_{0}}{{}^{a}H_{k}}({}^{a}\eta_{k} - {}^{a}\eta_{k-1}) + \beta(y - y_{0}); \; k = 1,3; \tag{5}$$

$$^{a}\eta_{k} = \frac{f_{0}}{{}^{a}g'_{k}}({}^{a}\psi_{k} - {}^{a}\psi_{k+1}),\; {}^{a}g'_{k} = g\frac{(\theta_{k+1} - \theta_{k})}{\bar{\theta}}; \; k = 1,2; {}^{a}\eta_{0} = {}^{a}\eta_{3} = 0. \tag{6}$$

Note that the layer indexing in the equations above goes from the surface upward and the Laplacian friction term is omitted; otherwise, these equations are completely analogous to the oceanic equations (the atmospheric variables are denoted above





by the left superscript 'a'). At the lower atmospheric boundary, the entrainment flux $^a e_0 = {}^a w_{ek}$ solely represents, in the

original Q-GCM formulation, Ekman dissipation (thus signifying the momentum transfer from the atmosphere to the ocean);

in the modified version of the model to be developed here it will also include a temperature dependent component capable of

driving mesoscale air–sea interaction (section 3.3). The atmospheric model (and thus the entire coupled model) is driven

through interior entrainment fluxes $^a e_1$, $^a e_2$ in Eq. (4) ($^a e_3$ is set to zero), which are a by-product of perturbing the mean-

state radiative-convective equilibrium (section 3.1) by a latitudinally non-uniform insolation. The new radiation/heat

exchange formulation and mixed layer/entrainment formulation developed here (section 3) are aimed to help achieve a

parameter regime with enhanced (and, arguably, more realistic) coupling between oceanic and atmospheric dynamics in the

model. They are further modified in formulating the new version of the model with an active hydrological cycle and the

associated latent-heat feedbacks (section 4).


## 3 Updates to the original, 'dry' version of Q-GCM

In describing the updates below, we will generally focus on the elements of Q-GCM model that have been revised here and

quote the values of new parameters, or the updated values of the previously used parameters along the way, while referring

the reader/user to the existing Q-GCM guide (Hogg et al., 2014) for a more thorough description of the default model

configuration.

### 3.1 Radiative-convective equilibrium, atmospheric mean state and convective fluxes

The previous version of Q-GCM assumes purely radiative equilibrium to compute the atmospheric mean state. In the revised

version, this assumption is replaced by that of the radiative-convective mean-state balance. We denote the actual (not

potential) vertically averaged temperatures within each of the interior atmospheric layers as $^a T_k$, $k = 1$, 2, 3 to write, over

ocean,

$$
\begin{aligned}
{}^o\rho\,{}^oC_p\,{}^oH_m\,{}^o\dot{T}_m &= -F_\lambda - F_0^\uparrow - F_m^\downarrow - F_s + {}^oF_m^{e+}, \\
{}^a\rho\,{}^aC_p\,{}^aH_m\,{}^a\dot{T}_m &= F_\lambda + F_0^\uparrow + F_m^\downarrow - \left(F_m^\uparrow + {}^aF_m^{e-} + F_1^\downarrow\right), \\
{}^a\rho\,{}^aC_p\,{}^aH_1\,{}^a\dot{T}_1 &= F_m^\uparrow + F_1^\downarrow + {}^aF_m^{e+} - \left(F_1^\uparrow + F_2^\downarrow + {}^aF_1^{e-}\right), \\
{}^a\rho\,{}^aC_p\,{}^aH_2\,{}^a\dot{T}_2 &= F_1^\uparrow + F_2^\downarrow + {}^aF_1^{e+} - \left(F_2^\uparrow + F_3^\downarrow + {}^aF_2^{e-}\right), \\
{}^a\rho\,{}^aC_p\,{}^aH_3\,{}^a\dot{T}_3 &= F_2^\uparrow + F_3^\downarrow + {}^aF_2^{e+} - F_3^\uparrow.
\end{aligned}
\tag{7}
$$

Here the dot denotes the time derivative and other notations follow the Q-GCM Users' Guide, v1.5.0 (Hogg et al. 2014). In

particular, the upward/downward arrow subscripts denote upwelling/downwelling longwave radiative fluxes within each of





the layers $k = m$ (mixed layer), 1, 2, 3 or the surface (0), the subscripts $e-$ and $e+$ denote entrainment fluxes below and above

interface $k$, the subscript $s$ refers to the solar radiation and $\lambda$ — to the ocean–atmosphere sensible/latent heat exchange. The fluxes describing non-radiative heat exchange between the layers are interpreted, in the mean state, as convective fluxes parameterized in the following way (cf. Manabe and Strickler 1964; Manabe and Wetherald 1967; Ramanathan and Coakley 1978):

$$
\begin{aligned}
\overline{F_\lambda} &= K({}^{o}\overline{T_m} - {}^{a}\overline{T_m} - \gamma_c \Delta H_m); \ \Delta H_m = {}^{a}H_m / 2, \\
{}^{a}\overline{F_m^e} &= {}^{a}\overline{F_m^{e-}} = {}^{a}\overline{F_m^{e+}} = K({}^{a}\overline{T_m} - {}^{a}\overline{T_1} - \gamma_c \Delta H_1); \ \Delta H_1 = \left({}^{a}H_m + {}^{a}H_1\right)/2, \\
{}^{a}\overline{F_1^e} &= {}^{a}\overline{F_1^{e-}} = {}^{a}\overline{F_1^{e+}} = K({}^{a}\overline{T_1} - {}^{a}\overline{T_2} - \gamma_c \Delta H_2); \ \Delta H_2 = \left({}^{a}H_1 + {}^{a}H_2\right)/2, \\
{}^{a}\overline{F_2^e} &= {}^{a}\overline{F_2^{e-}} = {}^{a}\overline{F_2^{e+}} = K({}^{a}\overline{T_2} - {}^{a}\overline{T_3} - \gamma_c \Delta H_3); \ \Delta H_3 = \left({}^{a}H_2 + {}^{a}H_3\right)/2.
\end{aligned}
\tag{8}
$$

The upward fluxes in Eqs. (8) are all positive (or are otherwise set to zero), with the coefficient $K = 200\,\text{W m}^{-2}\,\text{K}^{-1}$ and the critical lapse rate $\gamma_c = 6.5\,\text{K/km}$. The potential temperatures of the atmospheric layers are given by

$$
\begin{aligned}
\overline{\theta_m} &= {}^{a}\overline{T_m} + \Gamma_d\, {}^{a}H_m / 2, \\
\theta_1 &= {}^{a}\overline{T_1} + \Gamma_d \left({}^{a}H_m + {}^{a}H_1 / 2\right), \\
\theta_2 &= {}^{a}\overline{T_2} + \Gamma_d \left({}^{a}H_m + {}^{a}H_1 + {}^{a}H_2 / 2\right), \\
\theta_3 &= {}^{a}\overline{T_3} + \Gamma_d \left({}^{a}H_m + {}^{a}H_1 + {}^{a}H_2 + {}^{a}H_3 / 2\right),
\end{aligned}
\tag{9}
$$

where $\Gamma_d = g / {}^{a}C_p$ is the dry adiabatic lapse rate (of about 10 K/km).

The radiative fluxes in Eqs. (7) are parameterized assuming that the atmospheric layers have constant emissivity

$\varepsilon_m, \varepsilon_1, \varepsilon_2, \varepsilon_3$, and the Stefan-Boltzmann expressions for perturbation fluxes are linearized with respect to the basic-state:





$$
\begin{aligned}
F_0^\uparrow &= A_0 + D_0\, {}^oT_m', \\
F_m^\uparrow &= F_0^\uparrow(1-\varepsilon_m) + \varepsilon_m\left(A_m + B_m\, {}^aT_m'\right), \\
F_1^\uparrow &= F_m^\uparrow(1-\varepsilon_1) + \varepsilon_1\left(A_1 + B_1\, {}^aT_1'\right), \\
F_2^\uparrow &= F_1^\uparrow(1-\varepsilon_2) + \varepsilon_2\left(A_2 + B_2\, {}^aT_2'\right), \\
F_3^\uparrow &= F_2^\uparrow(1-\varepsilon_3) + \varepsilon_3\left(A_3 + B_3\, {}^aT_3'\right), \\
F_3^\downarrow &= -\varepsilon_3\left(A_3 + B_3\, {}^aT_3'\right), \\
F_2^\downarrow &= F_3^\downarrow(1-\varepsilon_2) - \varepsilon_2\left(A_2 + B_2\, {}^aT_2'\right), \\
F_1^\downarrow &= F_2^\downarrow(1-\varepsilon_1) - \varepsilon_1\left(A_1 + B_1\, {}^aT_1'\right), \\
F_m^\downarrow &= F_1^\downarrow(1-\varepsilon_m) - \varepsilon_m\left(A_m + B_m\, {}^aT_m'\right),
\end{aligned}
\tag{10}
$$

where

$$
\begin{aligned}
A_0 &= \sigma\, {}^o\overline{T}_m^{\,4}, \quad D_0 = 4\sigma\, {}^o\overline{T}_m^{\,3}, \\
A_m &= \sigma\, {}^a\overline{T}_m^{\,4}, \quad B_m = 4\sigma\, {}^a\overline{T}_m^{\,3}, \\
A_k &= \sigma\, {}^a\overline{T}_k^{\,4}, \quad B_k = 4\sigma\, {}^a\overline{T}_k^{\,3};
\end{aligned}
\tag{11}
$$

$k = 1, 2, 3$ and $\sigma$ is the Stefan–Boltzmann constant.

To solve for the mean state, we set all of $F_m^{e+}$, ${}^oT_m'$, ${}^aT_m'$, ${}^aT_1'$, ${}^aT_2'$, ${}^aT_3'$ to zero and numerically integrate Eqs. (7)–(11) to equilibrium, using Euler differences in time with the time step of 5 min. Setting $\varepsilon_m = \varepsilon_1 = \varepsilon_2 = \varepsilon_3 = 0.45$ along with $\overline{F}_s = -240$ W m$^{-2}$ results in the mean state whose parameters are listed in **Table 1**. The atmospheric optical depth decreases with altitude, but so do the unperturbed thicknesses of our chosen atmospheric layers, making the constant layer emissivities above a reasonable first approximation commensurate with an idealized nature of the present model. The model has a

realistic (time-mean, global-mean) vertical temperature distribution. Note that the atmospheric reduced gravities are derived, in this version of the model, from the mean-state parameters rather than being prescribed (at 1.2 and 0.4 ms$^{-2}$), as in the previous version (see the appendix for further details). The climatological solution above is formally obtained over ocean, but it also applies over land of zero heat capacity (since the steady state does not depend on the heat capacity of the surface); the land's zero heat capacity is also a feature of the original Q-GCM formulation. The near-surface convective fluxes,

however, would generally be different over ocean and over land (which occupies a significant fraction of the atmospheric channel, including fairly large strips both north and south of the ocean to avoid the distortion of ocean–atmosphere interaction by the effects associated with the atmospheric boundary conditions); the values of the convective fluxes in Table





1 should thus be interpreted to represent zonally averaged fluxes. Below, in section 3.2, we will describe, among other things, modifications of the atmospheric mixed layer (a.m.l.) perturbation equation (that is, the one describing evolution of

the anomalies with respect to the mean state) over land regions.

## 3.2 Mixed-layer perturbation equations and entrainment formulation

The perturbation equations in the mixed layers, for primed variables, have the same form as the first two equations (7), aside from addition of advective and entrainment fluxes which we will discuss further below; hereafter, we will drop primes in all

perturbation equations for convenience. We will assume that the atmospheric perturbation temperature is vertically uniform, consistent with an active role of convection processes (cf. Manabe and Wetherald, 1967), viz.

$$ {}^{a}T_{1}' = {}^{a}T_{2}' = {}^{a}T_{3}' \equiv {}^{a}T' = -({}^{a}\eta_{1}\Delta_{1}^{a}T + {}^{a}\eta_{2}\Delta_{2}^{a}T)/{}^{a}H; \quad {}^{a}H \equiv {}^{a}H_{1} + {}^{a}H_{2} + {}^{a}H_{3}, \tag{12} $$

($\Delta_{k}^{a}T$ above denotes the potential temperature jump across the $k$-th interface; see Table 1) which allows one to express all perturbation radiative fluxes in (10) via perturbation oceanic and atmospheric mixed layer temperatures ${}^{o}T_{m}$, ${}^{a}T_{m}$ and

interfacial displacements ${}^{a}\eta_{1}$, ${}^{a}\eta_{2}$ ; in particular,

$$
\begin{aligned}
F_{0}^{\uparrow} &= D_{0}\,{}^{o}T_{m}, \\
F_{m}^{\uparrow} &= D_{m}^{\uparrow}\,{}^{a}T_{m} + E_{0}^{\uparrow}\,{}^{o}T_{m}, \\
F_{1}^{\uparrow} &= A_{1,1}^{\uparrow}\,{}^{a}\eta_{1} + A_{1,2}^{\uparrow}\,{}^{a}\eta_{2} + D_{1}^{\uparrow}\,{}^{a}T_{m} + E_{1}^{\uparrow}\,{}^{o}T_{m}, \\
F_{2}^{\uparrow} &= A_{2,1}^{\uparrow}\,{}^{a}\eta_{1} + A_{2,2}^{\uparrow}\,{}^{a}\eta_{2} + D_{2}^{\uparrow}\,{}^{a}T_{m} + E_{2}^{\uparrow}\,{}^{o}T_{m}, \\
F_{3}^{\uparrow} &= A_{3,1}^{\uparrow}\,{}^{a}\eta_{1} + A_{3,2}^{\uparrow}\,{}^{a}\eta_{2} + D_{3}^{\uparrow}\,{}^{a}T_{m} + E_{3}^{\uparrow}\,{}^{o}T_{m}, \\
F_{3}^{\downarrow} &= A_{3,1}^{\downarrow}\,{}^{a}\eta_{1} + A_{3,2}^{\downarrow}\,{}^{a}\eta_{2}, \\
F_{2}^{\downarrow} &= A_{2,1}^{\downarrow}\,{}^{a}\eta_{1} + A_{2,2}^{\downarrow}\,{}^{a}\eta_{2}, \\
F_{1}^{\downarrow} &= A_{1,1}^{\downarrow}\,{}^{a}\eta_{1} + A_{1,2}^{\downarrow}\,{}^{a}\eta_{2}, \\
F_{m}^{\downarrow} &= F_{1}^{\downarrow}(1 - \varepsilon_{m}) + D_{m}^{\downarrow}\,{}^{a}T_{m},
\end{aligned}
\tag{13}
$$

where all of the $A$, $D$, $E$ coefficients can be written in terms of the known mean-state parameters. The equations (13) above should be compared with (4.2–4.6) of the Q-GCM User's Guide (Hogg et al. 2014). Notably, the parameter $\varepsilon_{m}$ was effectively set to 1 in the previous version of Q-GCM, and hence all of the $E$ coefficients were equal to zero. On the other

hand, that previous version had additional parameters $B$ and $C$ for radiative corrections associated with the variable a.m.l.





depth and topography. Here we are back to the model with a constant a.m.l. depth (see below); we also neglect topography corrections for simplicity.

Another consequence of the assumption $\varepsilon_m < 1$, used here, is that the coefficients $A$, $D$, $E$ in Eqs. (13) over ocean and over land are different, and so is the a.m.l. temperature equation — the second equation in (7). In particular, over land we have
(neglecting, for now, advection and entrainment terms in the a.m.l. equation)

$$
\begin{aligned}
0 &= -F_l^\uparrow - F_m^\downarrow - F_s, \\
{}^a\rho\,{}^aC_p\,{}^aH_m\,{}^a\dot{T}_m &= F_l^\uparrow + F_m^\downarrow - \left(F_m^\uparrow + {}^aF_m^{e-} + F_1^\downarrow\right),
\end{aligned}
\tag{14}
$$

where $F_l^\uparrow$ is the infrared upward flux from the surface of the land, the latter assumed to have zero heat capacity [hence zero on the left-hand side of the first equation in (14)] and conductivity (hence $F_\lambda = 0$). From Eqs. (14) it follows that over land

$$
\begin{aligned}
F_l^\uparrow &= -F_m^\downarrow - F_s, \\
{}^a\rho\,{}^aC_p\,{}^aH_m\,{}^a\dot{T}_m &= -F_s - \left(F_m^\uparrow + {}^aF_m^{e-} + F_1^\downarrow\right),
\end{aligned}
\tag{15}
$$

while, in analogy with Eqs. (10)

$$
F_m^\uparrow = F_l^\uparrow(1-\varepsilon_m) + D_m^\uparrow\,{}^aT_m = -(F_m^\downarrow + F_s)(1-\varepsilon_m) + D_m^\uparrow\,{}^aT_m
\tag{16}
$$

[compare this with the second Eq. (13)]. The first (additional) term in Eq. (16) will also modify all other upwelling radiation fluxes ( $F_1^\uparrow, F_2^\uparrow, F_3^\uparrow$ ) accordingly through Eqs. (10), resulting in modified values of the $A$ and $D$ coefficients and all $E$ coefficients set to zero over land.

Yet another, minor, but potentially fairly important modification of the previous Q-GCM formulation is the inclusion of the dependence on the relative wind speed $\left|{}^a\mathbf{u_m} - {}^o\mathbf{u_m}\right|$ in the bulk formulas for the sensible/latent heat ocean–atmosphere exchange, which plays a significant role in setting up the North Atlantic SST tripole variability (Deser and Blackmon, 1993; Kushnir, 1994; Czaja and Marshall, 2001; Kravtsov et al., 2007; Fan and Schneider, 2012):

$$
F_\lambda = \left(\lambda + {}^a\rho\,{}^aC_p\,C_h\left|{}^a\mathbf{u_m} - {}^o\mathbf{u_m}\right|\right)\left({}^oT_m - {}^aT_m\right)
\tag{17}
$$





[compare with Eqs. (4.7–4.9) in Hogg et al. 2014], where we use the values of $\lambda = 5$ W m$^{-2}$K$^{-1}$ and $C_h = 0.004$ ; with the

typical relative wind speed of $\left| {}^a\mathbf{u_m} - {}^o\mathbf{u_m} \right| \approx 7.5$ m s$^{-1}$ the magnitude of the total sensible/latent heat exchange coefficient in

Eq. (17) would be equal to 35 W m$^{-2}$K$^{-1}$, which is the value of $\lambda$ used in the previous edition of Q-GCM (Hogg et al.,

2014), along with the value of $C_h = 0$ .

To complete the mixed-layer heat conservation equations, we need to add advection, diffusion and entrainment heat fluxes,

namely:

$$
\begin{aligned}
&{}^o T_{mt} + \left( {}^o u_m \, {}^o T_m \right)_x + \left( {}^o v_m \, {}^o T_m \right)_y - \frac{{}^o w_{ek} \, {}^o T_m}{{}^o H_m} = {}^o K_2 \nabla_H^2 \, {}^o T_m - {}^o K_4 \nabla_H^4 \, {}^o T_m \\
&\hspace{4cm} + \frac{1}{{}^o \rho \, {}^o C_p \, {}^o H_m} \left( -F_\lambda - F_0^\uparrow - F_m^\downarrow - F_s + {}^o F_m^{e+} \right), \\
&{}^a T_{mt} + \left( {}^a u_m \, {}^a T_m \right)_x + \left( {}^a v_m \, {}^a T_m \right)_y + \frac{{}^a w_{ek} \, {}^a T_m}{{}^a H_m} = {}^a K_2 \nabla_H^2 \, {}^a T_m - {}^a K_4 \nabla_H^4 \, {}^a T_m \\
&\hspace{4cm} + \frac{1}{{}^a \rho \, {}^a C_p \, {}^a H_m} \left( F_\lambda + F_0^\uparrow + F_m^\downarrow - \left( F_m^\uparrow + {}^a F_m^{e-} + F_1^\downarrow \right) \right);
\end{aligned}
\tag{18}
$$

compare this with the first two equations (7) and with Eqs. (3.28–29) in Hogg et al. (2014). As mentioned above, in contrast

to the previous Q-GCM formulation, we use here the constant mixed-layer thickness in both the ocean and the atmosphere.

Therefore, in both the ocean and the atmosphere, entrainment is solely driven by the Ekman pumping. Neglecting vertical

diffusion and convection in the present perturbation model (which is another difference from Hogg et al., 2014), we write for

the ocean, following McDougall  and Dewar (1998),  Kravtsov et  al. (2007)

$$
\begin{aligned}
&{}^o F_m^{e+} = {}^o \rho \, {}^o C_p \, {}^o w_{ek} ({}^o T_1 - {}^o T_m), \quad {}^o F_m^{e-} = 0, \quad \text{if } {}^o w_{ek} > 0 \\
&{}^o F_m^{e+} = 0, \quad {}^o F_m^{e-} = -{}^o \rho \, {}^o C_p \, {}^o w_{ek} ({}^o T_1 - {}^o T_m), \quad \text{otherwise;}
\end{aligned}
\tag{19}
$$

cf. Eqs. (4.30–31) of Hogg et al. (2014). The entrainment in the ocean interior only occurs between layers 1 and 2, and is

computed in the same way as in the original model:


$$
{}^o e_1 = -\frac{{}^o F_m^{e-}}{\Delta_1^o T}
\tag{20}
$$





[Hogg et al. 2014, Eq. (4.32)]; following, again, Hogg et al. (2014), $^oe_1$ is also corrected to have zero area integral by adding a spatially uniform offset value at each time step.

Similarly, in the atmospheric mixed layer

$$
\begin{aligned}
&{}^aF_m^{e+} = {}^a\rho\,{}^aC_p\,{}^aw_{ek}({}^aT_m - {}^aT_1),\ {}^aF_m^{e-} = 0,\ \text{if}\ {}^aw_{ek} > 0\\
&{}^aF_m^{e+} = 0,\ {}^aF_m^{e-} = -{}^a\rho\,{}^aC_p\,{}^aw_{ek}({}^aT_m - {}^aT_1),\ \text{otherwise.}
\end{aligned}
\tag{21}
$$

Here $^aT_1$ is the perturbation temperature given by Eq. (12); in the mean state this temperature is set to $^a\overline{T_m}$. Using $^aT_1$ instead of $\theta_1$ in Eqs. (21) is what keeps the instability described by Hogg et al. (2003) in check in the present version of the model with constant $^aH_m$. This is due to the fact that $^aT_1$ is tied to the instantaneous vertical structure of the atmosphere, which limits the magnitude of entrainment heat fluxes (as $^aT_1$ tends to be closer to $^aT_m$ than $\theta_1$) and also provides additional negative feedbacks in the quasi-geostrophic potential vorticity (QGPV) equations via the dependence of entrainment fluxes 265    on $\eta$.

    Finally, a major modification in the present version of the Q-GCM model is the formulation of entrainment fluxes in the interior of the atmosphere. In the previous version of the model, all entrainment was assumed to occur at the lowest interface, leading to unrealistically small vertical shears of horizontal velocity in the upper atmosphere. Here we correct this by allowing the thermal forcing of the upper troposphere and entrainment through both atmospheric interfaces. The 270    perturbation heat conservation equations for the interior atmospheric layers can be obtained by setting the time derivatives on the left-hand side of the last three equations (7) to zero and using the jump conditions (McDougall and Dewar 1998) at the interfaces:

$$
\begin{aligned}
&0 = F_m^{\uparrow} + F_1^{\downarrow} + {}^aF_m^{e+} - \left(F_1^{\uparrow} + F_2^{\downarrow} + {}^aF_1^{e-}\right),\\
&0 = F_1^{\uparrow} + F_2^{\downarrow} + {}^aF_1^{e+} - \left(F_2^{\uparrow} + F_3^{\downarrow} + {}^aF_2^{e-}\right),\\
&0 = F_2^{\uparrow} + F_3^{\downarrow} + {}^aF_2^{e+} - F_3^{\uparrow},\\
&{}^aF_k^{e+} - {}^aF_k^{e-} = -{}^a\rho\,{}^aC_p\,{}^ae_k\Delta_k^aT;\ k = 1, 2
\end{aligned}
\tag{22}
$$

[compare this with Eqs. (4.10–4.12) of the Q-GCM User's guide; Hogg et al. (2014)]. Adding up the first three equations 275    (22) and using the fourth equation for the jump conditions allows one to write





$$\rho\,^a C_p \left(\,^a e_1 \Delta_1^a T + \,^a e_2 \Delta_2^a T\right) = F_m^{\uparrow} + F_1^{\downarrow} + \,^a F_m^{e+} - F_3^{\uparrow}. \tag{23}$$

Hogg et al. (2014) assumed $^a e_2 = 0$. We modify this assumption by making the entrainments across the lower and upper atmospheric interface be linearly related, with the coefficient $f_2$ (see below); this procedure can also be adapted for the use in an $n$-layer model by introducing additional free parameters analogous to $f_2$. To allow one a degree of freedom in controlling

damping rates at each interface somewhat independently, we also introduce here a (small) vertical diffusion, using a linearized version of McDougall and Dewar (1998) formulation:

$$
\begin{aligned}
^a e_1 &= \,^a e_1' + \mu_1^* \left( \frac{1}{\,^a H_1} + \frac{1}{\,^a H_2} \right) {}^a \eta_1 - \frac{\mu_1^*}{\,^a H_2}\,^a \eta_2, \\
^a e_2 &= \,^a e_2' + \mu_2^* \left( \frac{1}{\,^a H_2} + \frac{1}{\,^a H_3} \right) {}^a \eta_2 - \frac{\mu_2^*}{\,^a H_2}\,^a \eta_1, \\
^a e_2' &= f_2 \,^a e_1'.
\end{aligned}
\tag{24}
$$

We can now solve the system (23)–(24) for the two unknown non-diffusive entrainment rates $^a e_1'$ and $^a e_2'$ and, hence, for the full entrainment rates $^a e_1$ and $^a e_2$. We use the nominal value of $0.0001$ m s$^{-1}$ for both of $\mu_1^*$ and $\mu_2^*$ and initially set


$$f_2 = \frac{^a g_1'}{^a g_2'} \tag{25}$$

to ensure generation of similar velocity shears between atmospheric layers 1/2 and 2/3 by the thermal forcing of a given amplitude. Increasing $f_2$ would tend to increase the geostrophic zonal velocity shear between the lower two atmospheric layers and decrease the velocity shear between the upper two atmospheric layers; setting $f_2 = 0$ recovers the previous Q-GCM formulation. The optimal value of $f_2$ is to be determined by trial-and-error tuning of the model.

**3.3 Temperature-dependent flow in atmospheric mixed layer and partially coupled setup**

We introduce temperature dependence of the a.m.l. winds by modifying the mixed-layer momentum equations in two ways, namely: (i) including, explicitly, temperature driven pressure gradients (which takes into account the mixed-layer hydrostatic adjustment to temperature contrasts: Lindzen and Nigam, 1987), following Feliks et al. 2004, 2007, 2011; and (ii) making the surface drag coefficient depend on the ocean–atmosphere temperature difference to parameterize changes in a.m.l.

stability (Wallace et al., 1989), following Hogg et al. (2009); see Small et al. (2008) and Chelton and Xie (2010) for a review





of these two mechanisms for mesoscale air–sea coupling. Putrasahan et al. (2013) demonstrated that, in the Kuroshio region, both mechanisms (i) and (ii) are important, with relative contributions depending on the spatial scale of the SST anomalies. Putrasahan et al. (2017) also concluded that heat advection by oceanic mesoscale currents plays a key role in creating such SST anomalies and forcing the MABL response in the Gulf of Mexico. To implement these changes, we write the a.m.l.

momentum equations as

$$
{}^{a}u_{m} = {}^{a}u_{1} + \alpha_{T} \frac{1}{2} \frac{g\,{}^{a}H_{m}}{\theta_{0}f_{0}} \frac{\partial\,{}^{a}T_{m}}{\partial y} - \frac{{}^{a}\tau^{y}}{{}^{a}H_{m}f_{0}},
$$
$$
{}^{a}v_{m} = {}^{a}v_{1} - \alpha_{T} \frac{1}{2} \frac{g\,{}^{a}H_{m}}{\theta_{0}f_{0}} \frac{\partial\,{}^{a}T_{m}}{\partial x} + \frac{{}^{a}\tau^{x}}{{}^{a}H_{m}f_{0}}, \tag{26}
$$
$$
\left({}^{a}\tau^{x}, {}^{a}\tau^{y}\right) = C_{D}\max(1+\alpha\Delta T, 0.1)\left|{}^{a}\mathbf{u_{m}} - {}^{o}\mathbf{u_{m}}\right|\left({}^{a}u_{m} - {}^{o}u_{m}, {}^{a}v_{m} - {}^{o}v_{m}\right),
$$

where $({}^{a}u_{1}, {}^{a}v_{1})$ is the geostrophic velocity in the lowest atmospheric layer, $({}^{a}\tau^{x}, {}^{a}\tau^{y})$ is the wind stress, $\Delta T = {}^{o}T_{m} - {}^{a}T_{m}$, $\alpha_{T}$ =1 and $\alpha$ =0.15. Setting one of the $\alpha$-parameters to zero can be used to examine processes (i) and (ii) above independently; setting both of these parameters to zero would recover the previous, temperature-independent a.m.l. wind formulation (3.2)–

(3.3) of Hogg et al. (2014). On top of these modifications, we also set the drag coefficient over ocean to 2/3 of the default value over land, following Marshall and Molteni (1993).

Upon adding to (26) analogous equations for oceanic mixed layer in their original form [Hogg et al.  2014; Eqs. (3.4)–(3.5)], we end up with a closed system of equations for the unknown values of $({}^{a}\tau^{x}, {}^{a}\tau^{y})$ at each grid point, which can be solved analytically in the same way as before  (see Hogg et al., 2014). Note that additional temperature gradients in the

first two equations (26) produce a non-divergent wind field with zero direct Ekman pumping and, also, zero temperature advection contributions; their dynamical effect is thus purely indirect,  via modifications to the wind-stress field; they also generate non-zero moisture advection in the moist version of the Q-GCM model, to be developed later in section 4.

The temperature-dependent a.m.l. wind formulation (26) is associated with coupled feedbacks that tend to suppress oceanic turbulence and SST fronts (cf. Hogg et al., 2014; see also section 5). In principle, realistic mesoscale ocean field can

still be achieved in inherently more turbulent oceanic regimes at high Reynolds numbers, but this requires very high ocean resolution and is computationally demanding. An alternative fix is to apply partial momentum coupling of the oceanic and atmospheric mixed layers in which the atmosphere "sees" the wind stress as per the full version of Eqs. (26), whereas the oceanic wind stress is computed from Eqs. (26) in which $\alpha_{T} = \alpha = 0$. In this way the mesoscale feedbacks of temperature-dependent wind, which damp oceanic turbulence, are artificially suppressed, but their effect on the atmosphere is preserved,



possibly leading to coupled dynamics involving large-scale low-frequency reorganization of the wind field and the ensuing
ocean response.

### 3.4 Lateral boundary conditions for mass and temperature equations

The original Q-GCM formulation employed no-through-flow conditions on the zonal boundaries of the atmospheric channel
but effectively allowed the mass to leave/enter ocean mixed layer through side boundaries to avoid Ekman-pumping
singularities there [via the direct use of Eqs. (3.5) and (3.18) to compute the oceanic Ekman pumping in Hogg et al. (2014)];
this means, among other things, that the area integral of Ekman pumping over the ocean basin does not vanish. Although it
is a lesser problem in the atmospheric set up, we modify the computation of the atmospheric Ekman pumping at the zonal
boundaries accordingly to achieve a uniform model formulation and also to avoid an abnormal boundary pumping in the
atmosphere. To do so, instead of setting $^{a}v_m$ to zero at the zonal boundaries, we assign it the values computed by the second
Eq. (26), with the Ekman pumping computed as usual — in terms of the divergence of the mixed-layer horizontal velocity
field — by Eq. (3.16) or, equivalently, via curl of the wind stress by Eq. (3.17) in Hogg et al. (2014).

Naturally, with open boundaries in both the ocean and the atmosphere, we also let the fluid leaving/entering the basin to have
temperature determined by the Neumann boundary condition of zero temperature derivative in the direction normal to the
boundary:

$$\frac{\partial T_m}{\partial n} = 0, \tag{27}$$

where $T_m$ denotes either atmospheric or oceanic mixed-layer temperature. With the open boundary condition augmented by
Eq. (27), it is no longer necessary to specify the temperature at the ocean's equatorward boundary, as was done in Hogg et al.
(2014).

### 4 Moist version of the model: MQ-GCM

Perhaps the most important change to the original Q-GCM formulation is the inclusion of the hydrological cycle and latent-
heat feedbacks, resulting in what we refer to as the Moist Quasi-Geostrophic Coupled Model (MQ-GCM). Indeed, Czaja and
Blunt (2011) proposed that the oceans can influence the troposphere through moist convection over the regions with strong
mesoscale variability; see also Willison et al. (2013). To compute moisture variables in the model, we assume that the
vertical temperature profile at a given $(x, y)$ location is linear in $z$, temperature decreasing with altitude $z$ above the sea level
at the critical lapse rate $\gamma_c$:





$$T_k(z) = {}^a\overline{T}_m - (\Gamma_d - \gamma_c)\frac{{}^aH_m}{2} + {}^aT_k - \gamma_c z. \tag{28}$$

Here $T_k$ is the absolute temperature (in K) in layer $k$ [$k$ can be a symbol ($m$) when referring to the a.m.l. temperature or an index ($k$=1, 2, 3) when denoting the interior (quasi-geostrophic) layers of the atmospheric model], while ${}^aT_k$ in the interior are given by Eq. (12). In such a constant-lapse-rate atmosphere, the pressure $p(z)$ is related to temperature as

$$p(z) = p_0\left(\frac{T(z)}{T(0)}\right)^{\frac{g}{R\gamma_c}}, \tag{29}$$

where $g$ is the gravity acceleration and $R$ is the ideal gas constant for dry air. Combining the latter two equations and the ideal gas law $p = \rho R T$ at the basic state with ${}^aT_k = 0$, we can compute the representative densities of each layer by estimating them at the altitude $z$ corresponding to the mid-layer height (e.g., $z = {}^aH_m/2$ for the mixed layer, $z = {}^aH_m + {}^aH_1/2$ for layer 1, etc.); this gives, for the parameters in Table 1 and $p_0 = 10^5$ hPa, $(\rho_m, \rho_1, \rho_2, \rho_3) = (1.16, 1, 0.77, 0.52)$ kg m$^{-3}$. The saturation specific humidity $h_s$ is given by

$$h_s = \varepsilon\frac{e_s}{p}; \ e_s = e_0\exp\left(\frac{a(T - T_0)}{b(T - T_0) + T_r}\right), \tag{30}$$

where $\varepsilon = R/R_v = 0.62$ is the ratio of the dry-air and water-vapor gas constants and the saturation water vapor pressure $e_s$ is computed as in Bolton (1980), using the parameters $e_0 = 611.2$ hPa, $a = 17.67$, $b = 1$, $T_r = 243.5$ K and $T_0 = 273.15$ K. Given the a.m.l. perturbation temperature ${}^aT_m$ and the atmospheric interface displacements ${}^a\eta_1, {}^a\eta_2$, the equations (12), (28) –(30) can be used to compute the saturation specific humidity as a function of $z$ at every grid point ($x, y$) and within each atmospheric layer $k$= ($m$, 1, 2, 3).

The moist version of the Q-GCM model has, compared with the original dry model, additional variables representing the specific humidity $h_k(x, y, t)$ in each layer; these variables are discretized on the model's $T$-grid. The specific humidity is assumed to be independent of $z$ except when used in the formulas parameterizing moisture fluxes at the top and bottom of the a.m.l. (see below). The humidity equations in both the a.m.l. and the atmospheric interior are largely analogous to the a.m.l. temperature equation (18) (cf. Deremble et al. 2013) and are given by



$$h_{mt} + \left({}^a u_m h_m\right)_x + \left({}^a v_m h_m\right)_y + \frac{{}^a w_{ek} h_m}{{}^a H_m} = {}^a K_2 \nabla_H^2 h_m - {}^a K_4 \nabla_H^4 h_m + \frac{1}{\rho_m {}^a H_m}\left(E - P_m - F_m^{e-}\right);$$

$$h_{kt} + \left({}^a u_k^g h_k\right)_x + \left({}^a v_k^g h_k\right)_y = {}^a K_2 \nabla_H^2 h_k - {}^a K_4 \nabla_H^4 h_k + \frac{1}{\rho_k {}^a H_k}\left(F_{k-1}^{e+} - P_k - F_k^{e-}\right).$$

(31)

Here $\left({}^a u_k^g, {}^a v_k^g\right)$ are the geostrophic velocities in the atmospheric layer $k$, $E$ is the evaporation and $P_k$ is the precipitation;

$F_k^{e-}$ and $F_k^{e+}$ are the moisture entrainment fluxes below and above the interface $k$, respectively (all of these fluxes are in kg

m$^{-2}$ s$^{-1}$). Once again, $k= (m, 1, 2, 3)$ and $k-1 \to m$ for $k=1$. The equations (31) also use boundary conditions analogous to those for temperature (section 3.4).

The evaporation over the ocean is given by (Gill 1982)

$$\frac{E}{\rho_m} = C_E \left|{}^a \mathbf{u_m} - {}^o \mathbf{u_m}\right|\left(h_s\left({}^o \overline{T}_m + {}^o T_m\right) - h_{m,r}\right);$$

$$h_{m,r} = h_m \frac{h_s\left(T_m\left({}^a H_m / 2\right)\right)}{h_s\left(T_m(0)\right)},$$

(32)

where the atmospheric specific humidity near the ocean surface $h_{m,r}$ is computed assuming constant relative humidity in the

a.m.l., following Deremble et al. (2012); the coefficient $C_E = 1.5 \times 10^{-3}$. Over land, we specify the (fixed in time) evapotranspiration flux (which also includes the zonally averaged evaporation from other ocean basins absent in our one-basin configuration; this allows us to achieve reasonable values characterizing the moist model's climatological distribution of specific humidity). In space, this $y$-dependent flux decreases linearly from the $\dfrac{E}{\rho_w}$ maximum value of 1 m year$^{-1}$ at the

southern boundary of the atmospheric model (note the usage of water density $\rho_w$ =1000 kg m$^{-3}$ here, leading to precipitation estimates in terms of the equivalent water depth per unit time) to the minimum value of 0.1 m year$^{-1}$ at the northern boundary.

Entrainment fluxes of moisture in Eqs. (31) are formulated in a way analogous to the entrainment heat fluxes. In particular, at the top of the a.m.l.





$$F_m^{e+} = {}^a w_{ek}(\rho_m h_m - \rho_1 h_{1,r}), \; F_m^{e-} = 0, \; \text{if} \; {}^a w_{ek} > 0$$
$$F_m^{e+} = 0, \; F_m^{e-} = -{}^a w_{ek}(\rho_m h_m - \rho_1 h_{1,r}), \; \text{otherwise;} \tag{33}$$
$$h_{1,r} = h_1 \frac{h_s\left(T_1({}^a H_m)\right)}{h_s\left(T_1({}^a H_m + {}^a H_1/2)\right)};$$


compare with (21). Here, the layer-1 reference specific humidity just above the a.m.l. $h_{1,r}$ is computed by assuming constant relative humidity in layer 1 [cf. Deremble et al. (2012) and Eqs. (32)]. Entrainment moisture fluxes in the geostrophic interior ($k$=1, 2) are given by

$$F_k^{e+} = {}^a e_k(\rho_k h_k - \rho_{k+1} h_{k+1}), \; F_k^{e-} = 0, \; \text{if} \; {}^a w_{ek} > 0$$
$$F_k^{e+} = 0, \; F_k^{e-} = -{}^a e_k(\rho_k h_k - \rho_{k+1} h_{k+1}), \; \text{otherwise,} \tag{34}$$

with $F_3^{e-}$ =0; the entrainment rates ${}^a e_k$ are computed using *the original formulas* (23), (24) of the *dry* Q-GCM model (see further discussion below).

The precipitation rates are computed following Laîné et al. (2011) methodology, except for using the linear (instead of Laîné et al.'s quadratic) local atmospheric temperature profiles Eq. (28). In particular, the moisture equations (31) are first stepped forward with all the precipitation rates set to zero to update the values of the specific humidity; recall that the specific 395    humidity is assumed to be independent of $z$ in each layer. Then the vertical integrals of the newly computed specific humidity excess over the saturated specific humidity (which is a function of $z$) within each layer are computed [for numerical efficiency, this is done semi-analytically by fitting a quadratic function of $z$ to $h_k-h_s(z)$]. This amount of moisture is set to fall out, over the period $2\Delta^a t$ associated with the leap-frog time step, as the precipitation $P_k$, and the specific humidity in the corresponding layer is reduced accordingly.

The hydrological cycle above is coupled with the model dynamics via the associated latent heat exchange/release. In the MQ-GCM, the equation (17) is only meant to describe the sensible heat exchange between the ocean and the atmosphere, with a reduced value of the sensible-heat exchange coefficient $C_h = 7\times10^{-4}$. In addition, the oceanic mixed layer is experiencing the (perturbation) latent heat loss (in W m$^{-2}$) of

$${}^o Q_L = L\left(E - \langle E \rangle\right) \tag{35}$$

and the atmospheric layers — the (perturbation) latent heat gain of





$$^{a}Q_{L,k} = L\left(P_k - \left\langle P_k \right\rangle\right), \tag{36}$$

where $L = 2.5 \times 10^6$ J kg$^{-1}$ is the latent heat of vaporization of water and $k = (m, 1, 2, 3)$. Note that the full latent-heat fluxes $LE, L P_k$ both include the part associated with the basic state of the model in its radiative–convective balance, but the Q-GCM is formulated as a perturbation model, which requires the subtraction of the basic-state latent-heat fluxes. We here

assume that the basic-state part of $^{o}Q_L$ and $^{a}Q_{L,k}$ is approximately given by the spatial averages of $LE, L P_k$ over the oceanic basin and atmospheric channel, respectively (this assumption is justified *post hoc* by the moist model's a.m.l. and o.m.l. climatological temperatures being close to those of a dry model) — $\left\langle E \right\rangle$ and $\left\langle P_k \right\rangle$; hence, we remove these spatial averages at each time step in Eqs. (35) and (36) to define the latent-heat flux anomalies that force our perturbation heat equations.

The fluxes $^{o}Q_L$ and $^{a}Q_{L,m}$ directly enter the right-hand side of the o.m.l. and a.m.l. equations (18), respectively. The

interior latent-heat release $^{a}Q_{L,k}$ is added to the right-hand side of the corresponding layer's heat equation in Eqs. (22), so that the sum $\sum_{k=1}^{3} {}^{a}Q_{L,k}$ enters, as an additional term, the right-hand side of Eq. (23) and modifies the entrainment rates $^{a}e_k$. Note again, however, that the moisture equations (31) use the first-guess, unmodified ("dry") entrainment rates computed from the original Eqs. (23), (24), upon which the precipitation rates $P_k$ and latent heat corrections $^{a}Q_{L,k}$ are computed and used to adjust the entrainment rates as follows

$$^{a}e_1 \rightarrow {}^{a}e_1 + \Delta {}^{a}e_1; \quad e_2 \rightarrow {}^{a}e_2 + f_2 \Delta {}^{a}e_1;$$
$$\Delta {}^{a}e_1 = \frac{\sum_{k=1}^{3} {}^{a}Q_{L,k}}{\rho {}^{a}C_p \left(\Delta_1^{a} T + f_2 \Delta_2^{a} T\right)}. \tag{37}$$

These modified entrainment rates are then used to timestep the atmospheric QGPV equations.

The above numerical scheme — with the first guess 'dry' entrainment driving the moisture equations to produce latent-heat-driven corrections to the entrainment, which are then used to force the QG model interior, — can be further improved by iterating the solution of the moisture equations at a given time step to achieve mutually consistent estimates of both

precipitation and entrainment in the interior QG layers. In this scheme, the interior entrainment fluxes at a given iteration would be used, along with the fixed advective and diffusive fluxes, to update the interior humidity and compute the precipitation rates until these rates (and entrainment rates) converge to a steady solution. This procedure is, however, much more numerically challenging than its first guess 'dry entrainment' implementation. The latter 'dry entrainment'





implementation may formally be justified if the "moist" corrections to the dry entrainment produce relatively small changes
to the interior precipitation. To explore this issue further, we included, in the present version of MQ-GCM model, an option
which allows one to modify the 'dry' entrainment based precipitation estimates via one additional iteration in which the
interior moisture equations (31) are stepped with the entrainment moisture fluxes (34) utilizing the moisture corrected
entrainment rates (37).

Moisture transport and latent heat release driving the atmospheric response in the areas away from the oceanic warm regions
of evaporation are important elements of air–sea interaction over variable SST fronts, which are altogether missing in a dry
version of the Q-GCM model (cf. Deremble et al. 2012).

## 5 Model simulations

We set the amplitude of the variable incoming radiation $| F_s' |$ to 150 W m$^{-2}$ (as compared to 80 W m$^{-2}$ in Hogg et al.,
2014) and ran three 130-yr simulations of the new dry version of the model, as well as of the new moist model, MQ-GCM
(six simulations total), using model setups with and without temperature dependence in the atmospheric mixed-layer wind.
For both dry and moist versions of the model, the control run, without this dependence, was started from the final state of the
preliminary 100-yr spin-up simulation (from rest), the partially coupled simulation was initialized by the final state of the
control run, and the fully coupled temperature-dependent simulation continued from the final state of the partially coupled
run. We disregarded the first 30 years to allow for model spin-up and adjustment (chosen based on the ocean energy
diagnostic) and analysed the last 100 yr of each simulation. Below we describe the results from the moist model runs only; in
the present parameter regime, the qualitative and quantitative behaviour of the companion dry model turned out to be very
similar and hence the dry-model results are not shown here (see section 6 for further discussion).

The atmospheric mean state does not appear to depend substantially on whether the temperature feedback on a.m.l. wind is
included in the model or not. In each case the atmosphere is characterized by a straight climatological jet with a reasonable
vertical shear (**Fig. 1**) and some zonal modulation; in particular, surface winds tend to be a bit stronger over ocean (due to
reduced surface drag), but barotropic wind exhibits an opposite modulation (weaker wind over ocean), consistent with
reduced temperature gradient over ocean (**Fig. 2**). On top of this mean state, the atmosphere is characterized by a vigorous
synoptic turbulence (**Fig. 3**).

The time-mean ocean currents (**Fig. 4**) represent large-scale subtropical and subpolar gyres and strong inertial recirculations,
which help maintain an intense eastward jet in the control and partially coupled simulations. The inertial recirculations
largely collapse in the fully coupled simulation (see, for example, the discussion in section 3 and Hogg et al., 2009), with
barotropic transports there (~40 Sv) only about 1/3 of those in the control and the partially coupled runs. Accordingly, the





eastward jet becomes much weaker and so is the climatological SST front (**Fig. 5**); this has probably much to do with anomalous Ekman pumping structures over the inertial gyres seen in the fully coupled simulation (Fig. 5, bottom right). The
ocean resolution (of 10 km) requires a relatively high viscosity (and low Reynolds number), and the ocean state can probably be characterized as eddy-permitting, rather than eddy resolving here (**Fig. 6**). The current simulations are only meant to illustrate a qualitative performance of the model. Simulations at higher resolutions (5 km) and Reynolds numbers may be advisable in all cases (cf. Martin et al. 2020).

We now show the moist characteristics of the model in the control simulation with 'dry entrainment' formulation (23)–(24)
of moisture entrainment fluxes in Eq. (34). **Figure 7** displays a segment of the basin-mean specific humidity time series for all of the atmospheric layers, featuring a reasonable vertical distribution of this quantity. The basin-mean moisture budget time series are shown in **Fig. 8**. The net evaporation/precipitation in the model are both around 0.6 m yr$^{-1}$, which is lower than the observed global-mean values of about 1 m yr$^{-1}$ due to the lack of tropical dynamics in the QG formulation. The basin-mean $E$–$P$ is slightly unbalanced, indicating a small moisture flux of about 0.1 m yr$^{-1}$ through the open boundaries of
the model (see sections 3.4 and 4). The specific humidity climatological distributions (**Fig. 9**) are slightly zonally nonuniform due to land–sea contrast and have patterns generally consistent with the atmospheric temperature distributions in Fig. 2.

The evaporation (**Fig. 10**, left) is prescribed and zonally uniform over land (see section 4) but is active over ocean, exhibiting reduced values to the north and enhancement along the southern and western boundary of the ocean and the double-gyre
confluence zone. Atmospheric boundary-layer precipitation is also enhanced over these areas (Fig. 10, right) and exhibits relative minima over the rest of the ocean. Globally, precipitation reaches local maximum at the southern boundary of the model and global minimum at the northern boundary and has a dipolar zonal structure around the axis of the channel, with precipitation minimum/maximum at the anticyclonic(equatorward)/cyclonic(poleward) flanks of the mid-latitude jet, respectively. These features also translate, to some extent, to the precipitation distribution in the atmospheric interior (**Fig.
11**), although land–sea contrast in precipitation in the interior is opposite in sign to the one within the atmospheric boundary layer. The climatological distribution of precipitation is the result of averaging over intermittent, in space and time, precipitation episodes, as illustrated by a snapshot example in **Fig. 12**.

Finally, we present here initial evidence for an important effect of the temperature-dependent wind-stress formulation on the low-frequency dynamics of MQ-GCM. This effect can be noticed in the behaviour of the leading EOF of SST (**Fig. 13**),
which is dominated, in all simulations, by a monopolar (in $y$) SST pattern in the region of ocean's eastward jet and its extension. The intensity, meridional localization and west-to-east scale of this pattern are the largest in the partially coupled run, which has the strongest oceanic turbulence capable of affecting mesoscale winds above the oceanic eastward jet (Fig. 13, top middle panel). This variability tends to be suppressed in both the control run (with no direct SST effect on the a.m.l.



winds: Fig. 13, top left panel) and the fully coupled run with active mesoscale coupling (but in which the ocean eddies are
partly suppressed: Fig. 13, top right panel), indicating the importance of both the ocean eddies and air–sea mesoscale
coupling for this mode. Furthermore, this mode's time-dependence is characterized by a pronounced interdecadal variability
in the partially coupled simulation, whereas the dominant time scales in the control and fully coupled simulations are shorter
(interannual-to-decadal) and the associated variances — smaller (Figs. 13, bottom row), with the energy-density ratio of the
partially coupled to control run of about 6 at low frequencies. It is not immediately clear, however, whether this mode
imprints itself onto the atmosphere even in the partially coupled simulation. There are indications that this run's leading jet-
shifting EOF of the atmospheric streamfunction (analogous to that of the control run shown in the top panel of **Fig. 14**) has
an enhanced energy at the low-frequency end of the spectrum compared to the control and fully coupled simulations (Fig.
14, bottom), but this enhancement is not statistically significant and may be due to sampling. Longer and — most
importantly — more highly resolved simulations, in both the ocean and the atmosphere (cf. Martin et al. 2020), are required
to gauge the potential of the active mesoscale air–sea coupling to fuel decadal climate modes.

## 6 Discussion and conclusions

Note that the dynamical core of the present MQ-GCM model is identical to that of the original Q-GCM model, which has
already been used in a suite of studies addressing mid-latitude climate variability, while the newly added physics elements
have either been tested and verified in ocean-only or atmosphere-only settings (e.g., temperature dependent wind stress:
Feliks et al. 2004, 2007; Hogg et al., 2009) or are largely analogous, in their numerical formulation, to the previous Q-GCM
elements (e.g., moisture advection and time stepping scheme is analogous to that of the mixed-layer temperature), or are
borrowed from similar ocean–only or coupled models (e.g., evaporation and latent heat exchange parameterizations borrow
from Kravtsov et al. 2006 and Deremble et al. 2012, 2013). The novelty of the present model is in that all these elements are
brought together in a fully coupled setting, which makes it a unique numerically efficient tool for exploring possible
dynamics of the midlatitude coupled climate variability. While of intermediate complexity, the model is still fairly involved
and no reference analytical solutions to directly verify the accuracy of its numerical implementation are available.
Furthermore, it is well known from decades of numerical experimentation with analogous ocean-only models (see, e.g.,
section 1 and references therein) that the regimes of simulated geostrophic turbulence strongly depend on the model's
effective Reynolds number, with larger values of this number achievable at a higher horizontal resolution. An analogous
statement pertains to frontal air–sea interactions, which may be sensitive to the horizontal resolution in the atmospheric
component of the model (see Feliks et al. 2004, 2007). Future studies of the model sensitivity to the horizontal resolution, in
the parameter regimes with high Reynolds numbers and mesoscale-resolving atmosphere are perhaps the ones to bring about
the most interesting dynamical insights.

In the present model development effort, we only utilized eddy-permitting ocean resolution and a coarse-grid atmospheric
model, which may be behind the similarity between the dry-model and moist-model behaviour noted in section 5. Overall,





the MQ-GCM model exhibits rich moisture dynamics reminiscent of many of the observed properties of the Earth's climate system. It remains to be seen whether these and other processes (such as mesoscale air–sea coupling) affect in fundamental ways the dynamics of the mid-latitude ocean–atmosphere system's coupled decadal variability. Preliminary results above indicate that the model's low-frequency variability is indeed sensitive to the details of air–sea interaction; furthermore, both

dry and moist versions of the atmospheric model — in parameter ranges corresponding to strong thermal driving and intermediate surface drag (e.g., $C_D = 0.0005$) — now exhibit bimodality of the type documented in Kravtsov et al. (2005, 2006, 2007, 2008), which is likely to be important for any decadal climate modes supported by the model (not shown here).

**7 Summary of model updates and code modifications**

Changes to the model formulation are summarized below in **Table 2**. **Table 3** outlines the corresponding changes in the Q-GCM source code.

**Appendix: QGPV equation in a layered atmospheric model**

Consider an ideal-gas dry atmosphere comprised of layers with constant potential temperatures $\theta_k$. Using the definition of

potential temperature, ideal gas law, assuming hydrostatic balance and dropping, in this section, the left superscript "a" that denotes atmospheric variables in the main text —

$$\theta = T\left(\frac{P_0}{P}\right)^{\frac{R}{C_p}}; \; P = \rho R T; \; \frac{\partial P}{\partial z} = -\rho g \tag{A1}$$

—    one can express the pressures within each layer $P_k$ as

$$P_k^{\frac{R}{C_p}} = -P_0^{\frac{R}{C_p}}\frac{z}{H_{\theta k}} + C_k(x,y); \; H_{\theta k} \equiv \frac{C_p \theta_k}{g}. \tag{A2}$$

The perturbation fields $C_k$ can be found by requiring the pressure to be continuous across each atmospheric interface, namely

$$P_1\big|_{H_m} = P_m(x,y,t); \; P_1\big|_{H_m + H_1 + \eta_1} = P_2\big|_{H_m + H_1 + \eta_1}; \; P_2\big|_{H_m + H_1 + H_2 + \eta_2} = P_3\big|_{H_m + H_1 + H_2 + \eta_2}, \tag{A3}$$

where $P_m$ is the pressure at the top of the atmospheric mixed layer. For example, from (A1)–(A3) we have, for the first atmospheric layer

$$T_1 = \theta_1 F_1; \; P_1 = P_0 F_1^{\frac{C_p}{R}}; \; \rho_1 = \frac{P_0}{R\theta_1}F_1^{\frac{C_p}{R}-1}, \tag{A4a}$$





where

$$F_1(x,y,z,t) = \left(\frac{P_m}{P_0}\right)^{\frac{R}{C_p}} - \frac{(z-H_m)}{H_{\theta 1}}.$$

(A4b)

From (A4), the horizontal pressure gradient force in this layer is

$$-\frac{1}{\rho_1}\nabla P_1 = -\nabla C_p \theta_1 \left(\frac{P_m}{P_0}\right)^{\frac{R}{C_p}} = -\nabla \psi_1 f_0.$$

(A5a)

Here, the lower layer streamfunction $\psi_1$ is given by

$$\psi_1 = \frac{C_p \theta_1}{f_0}\left(\frac{P_m}{P_0}\right)^{\frac{R}{C_p}} \approx \frac{C_p \overline{\theta}}{f_0}\left(\frac{P_m}{P_0}\right)^{\frac{R}{C_p}},$$

(A5b)

where $\overline{\theta} \sim 300$ K is the representative atmospheric potential temperature taken here to be equal, approximately, to the vertical average of individual-layer potential temperatures. In an analogous way, we find, for streamfunctions in layers 2 and 3

$$\psi_2 = \frac{C_p \theta_2}{f_0}\left(\left(\frac{P_m}{P_0}\right)^{\frac{R}{C_p}} - \left(\frac{1}{H_{\theta 1}} - \frac{1}{H_{\theta 2}}\right)\eta_1\right) \approx \psi_1 - \frac{g_1' \eta_1}{f_0},$$

(A5c)

$$\psi_3 = \frac{C_p \theta_3}{f_0}\left(\left(\frac{P_m}{P_0}\right)^{\frac{R}{C_p}} - \left(\frac{1}{H_{\theta 1}} - \frac{1}{H_{\theta 2}}\right)\eta_1 - \left(\frac{1}{H_{\theta 2}} - \frac{1}{H_{\theta 3}}\right)\eta_2\right) \approx \psi_2 - \frac{g_2' \eta_2}{f_0},$$

(A5d)

where we assumed that the differences between the potential temperatures of individual layers are small compared to $\overline{\theta}$ and estimated reduced gravities as

$$g_k' = g\frac{\theta_{k+1} - \theta_k}{\overline{\theta}}.$$

(A6)

From (A5) it follows that

$$\eta_k = f_0 \frac{\psi_k - \psi_{k+1}}{g_k'}$$

(A7)

and that the "dynamic pressures" $^a p_k$ in Hogg et al. (2014) are equal to $f_0 \psi_k$.

Furthermore, under synoptic scaling, the continuity equation in each layer is approximately

$$\frac{\partial u}{\partial x} + \frac{\partial v}{\partial y} + \frac{\partial w}{\partial z} - \frac{w}{H_{\theta k}} = 0,$$

(A8)



but the last term on the right-hand side of (A8) — approximating $\dfrac{w}{\rho}\dfrac{\partial \rho}{\partial z}$ — is smaller than other terms and can be neglected

under QG scaling, since $H_{\theta k} \simeq 30\,\mathrm{km}$ and $H_k / H_{\theta k} \sim 0.1$ is on the order of the Rossby number.

Thus, the above scaling arguments and calculations demonstrate the validity of Boussinesq approximation for quasi-geostrophic compressible atmosphere and justify the uniform treatment of oceanic and atmospheric dynamics in Hogg et al. (2014).


**Code availability**

The updated code alongside basic instructions on its use (see readme file there), as well as restart files for the six simulations described in this paper, are publicly available from GitHub at https://github.com/GFDANU/q-gcm (Kravtsov et al. 2021).

To use the code, one should replace the routines and scripts of the original source code (publicly available at http://www.q-gcm.org with the reference to https://github.com/GFDANU/q-gcm) summarized in Table 3 by their updated versions (contained in the new MQ-GCM folder on GitHub) and compile/run the resulting executable in the same way as before (see Hogg et al., 2014).

**Author contribution**

SK initiated the study, led the development of MQ-GCM modifications, ran some of the numerical experiments and produced the initial draft of the paper; IM performed and analysed coupled and uncoupled experiments with and without mesoscale air–sea coupling using the original and updated versions of Q-GCM; all co-authors contributed to MQ-GCM development, to analysing the numerical simulations and to writing of the manuscript. We dedicate this paper to the memory of our friend and colleague Prof. Peter Killworth, one of the developers of the original Q-GCM model.

**Competing interests**


The authors declare that they have no conflict of interest.

**Acknowledgements**

Initial stages of this research were funded through the 2018 University of Wisconsin-Milwaukee Research Growth Initiative (RGI) program (SK, IM). SK also acknowledges partial support from the Russian Science Foundation (Contract 18-12-590 00231) (model development and theoretical considerations; sections 1–3) and from the Russian Ministry of Education and Science (project 14.W03.31.0006) (numerical experiments and interpretation of the results; sections 3–6). WKD was supported by NSF grants OCE-1829856 and OCE-2023585, as well as by the CNRS/ANR grant ANR-18-MPGA-0002.





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





**Table 1:** **Mean-state parameters derived from (7)–(11), except for the last two rows detailing the ocean mean state based, loosely, on the observed oceanic vertical structure (note the difference here with the values used in the previous Q-GCM version).**


| Parameters | Value(s) | Description, units |
|---|---|---|
| $\left( {}^{o}\overline{T_m},\ {}^{a}\overline{T_m},\ {}^{a}\overline{T_1},\ {}^{a}\overline{T_2},\ {}^{a}\overline{T_3} \right)$ | (286.5, 282.6, 272.4, 255.9, 233.1) | Atmospheric mean-temperature structure ($K$) |
| $\left( {}^{o}\overline{T_m},\ {}^{a}\overline{\theta_m},\ \theta_1,\ \theta_2,\ \theta_3 \right)$ | (286.5, 287.6, 292.4, 300.9, 313.1) | Atmospheric potential-temperature structure ($K$) |
| $\Delta_1^a T \equiv \theta_2 - \theta_1,\ \Delta_2^a T \equiv \theta_3 - \theta_2$ | 8.5,   12.2 | Cross-interfacial temperature difference ($K$), atmosphere |
| $\overline{\theta}$ | 302.7 | Vertically averaged mean potential temperature ($K$) |
| ${}^{a}g'_k = g\Delta_1^a T / \overline{\theta}$ | (0.3, 0.4) | Reduced gravity (m s$^{-2}$), atmosphere |
| $\left( \overline{F_\lambda},\ {}^{a}\overline{F_m^e},\ {}^{a}\overline{F_1^e},\ {}^{a}\overline{F_2^e} \right)a$ | (143.3, 90.5, 45.3, 15.8) | Convective heat fluxes (W m$^{-2}$) |
| $\left( {}^{o}T_1, {}^{o}T_2, {}^{o}T_3 \right)$ | ${}^{o}\overline{T_m} - (2, 10, 14)$ | Oceanic mean-temperature structure ($K$) |
| $\Delta_1^o T \equiv {}^{o}T_1 - {}^{o}T_2,\ \Delta_2^o T \equiv {}^{o}T_2 - {}^{o}T_3$ | 8, 4 | Cross-interfacial temperature difference ($K$), ocean |






**Table 2. Differences between updated and original Q-GCM formulation.**

| Updates | Original model |
|---|---|
| New radiative-convective scheme to derive the atmospheric mean state and perturbation equations | Purely radiative equilibrium |
| Constant thickness, gray-body atmospheric mixed layer | Variable thickness, blackbody atmospheric mixed layer |
| Reduced gravities (and, hence, Rossby radii) in the atmosphere computed using the mean-state parameters derived from the radiative-convective equilibrium above | Reduced gravities (and Rossby radii) specified independently of the radiative-equilibrium mean-state parameters |
| Modified entrainment parameterization: <br><br> • Ekman driven, upwind differencing in the oceanic mixed layer <br> • Ekman driven, upwind differencing in the atmospheric mixed layer, with effective interior temperature depending on atmospheric state, resulting in stable formulation <br> • Interior entrainment active in both lower and upper atmosphere, optional addition of weak vertical diffusion | <br><br> • Ekman driven, central differences <br><br> • Turbulence driven, depending on variable mixed-layer thickness, with zero contribution to interior thermal forcing to achieve stability <br><br> • Interior entrainment in the lower atmosphere only |
| Sensible/latent heat exchange with (relative) wind-speed dependence in the bulk formulas | No dependence on wind speed in the bulk formulas |
| Reduced surface drag over ocean | Uniform surface drag |
| Convection is neglected | Convective adjustment in both mixed layers |
| Temperature dependent mixed-layer winds, with a possibility of a partially coupled set up, in which ocean and atmosphere experience different wind stress | Mixed-layer winds that do not depend on temperature |
| Open boundary conditions for mass and temperature equations in the mixed-layer formulation for both ocean and atmosphere | No-flow (for atmosphere only) insulating conditions (for both mixed layers), except for the specified temperature at the oceanic mixed layer's southern boundary (in the Northern Hemisphere formulation) |
| Addition of an active hydrological cycle, including moisture advection and latent heat feedbacks in the atmosphere | Dry model |







**Table 3. Changes in the source code.**

| File name | Summary of changes |
|---|---|
| Makefile | modified to account for new hra_out (high-resolution atmospheric output) module, as well as for new dependencies between modules |
| make.config | includes new model options using flags **highres_output** (high-resolution atmospheric output over ocean), **temp_fdbck** (temperature dependent a.m.l. winds) and **partial_coup** (partial temperature-dependent wind-stress coupling). Note: do not activate **sb_hflux** or **nb_hflux** options due to new — open boundary — conditions in ocean mixed layer. The new moist model formulation is activated by the flag **moist**. The option **dry_latent** uses the 'dry entrainment' estimates (23)–(24) in computing the moisture entrainment fluxes in Eq. (34); without this flag, the updated estimates as per Eq. (37) are used |
| input.params | the spaces previously used for optical depths are now used for the corresponding layer emissivity, stored in zm, zopt($k$); note that the atmospheric and oceanic mean-state parameters, including atmospheric reduced gravities, are now overwritten in radsubs.f. For the moist model: several new parameters are added, in particular the parameters $C_h$ and $C_e$, new output options are included as well |
| out_param.F | modified to output additional parameters associated with the new radiative-convective scheme and the moist-model formulation |
| parameters_data.F | added parameters related to high-resolution atmospheric output over ocean; moved the placement of the ocean off the axis of the atmospheric channel to break north–south symmetry |
| atstate_data.F | added workspace for entrainment across the upper atmospheric interface entat1 |
| atconst_data.F | For the moist model, added atmospheric-layer representative densities |
| intrfac_data.F | added ssta: the atmospheric-resolution SST field over ocean obtained by averaging ocean-resolution SST within the boundaries of atmospheric cells; for the moist model: added evapa (evaporation at atmo. resolution), new heat-exchange coefficient variables $C_e$ and $C_h$ |
| radiate_data.F | modified to include parameters of entrainment formulation across both atmospheric interfaces (rather than only for the lower interface, as before) |
| hra_out.F, highresout.F | new module/additional code performing high-resolution atmospheric output over ocean |
| radsubs.F | new radiative-convective scheme; for the moist model: new moist-model parameters: initial humidities, atmospheric-layer densities, initialization of evaporative flux over land |
| xfosubs.F | xforc: temperature-dependent a.m.l. wind (involving, among other things, an additional bicubic interpolation of a.m.l. temperature to ocean resolution), partial coupling option, updated formulation of a.m.l. and o.m.l. forcing, including wind-speed |





| | |
|---|---|
| | dependence of the sensible/latent heat exchange. The parameters $\alpha_T$ and $\alpha$ are specified within xforc and can be zeroed out individually to test different mechanisms of temperature dependent wind. Code in bcuini modified to compute the actual (non-zero) wind stress at the atmospheric zonal boundaries, with an option (in comments) to close the boundary by imposing the no-through-flow condition. For the moist model: modified formulation of the sensible/latent heat exchange. |
| amlsubs.F | aml: constant a.m.l. thickness,  new radiation and entrainment model, convection neglected;  amladf: open boundaries for advection. For the moist model: step atmospheric humidity equations, modify latent-heat corrections to the interior entrainment. |
| omlsubs.F | oml: upwind advection for Ekman-driven entrainment (in the current form, the o.m.l. entrainment is set to use the climatological difference between o.m.l. and layer-1 temperature: this results in a bit larger and more realistic SST north–south SST contrasts); omladf: open boundaries for advection. |
| qgasubs.F | added PV sources due to entrainment in the upper atmosphere (across the upper interface) |
| q-gcm.F | modified  to include high-atmo-resolution output module;  call  to radiat  moved up to update and record the mean-state parameters computed and overwritten by radsubs.f (these are: atmospheric and oceanic mean temperatures, atmospheric reduced gravities). For the moist model: new I/O specifications and initialization. |
| nc_subs.F | For the moist model: new netcdf I/O |








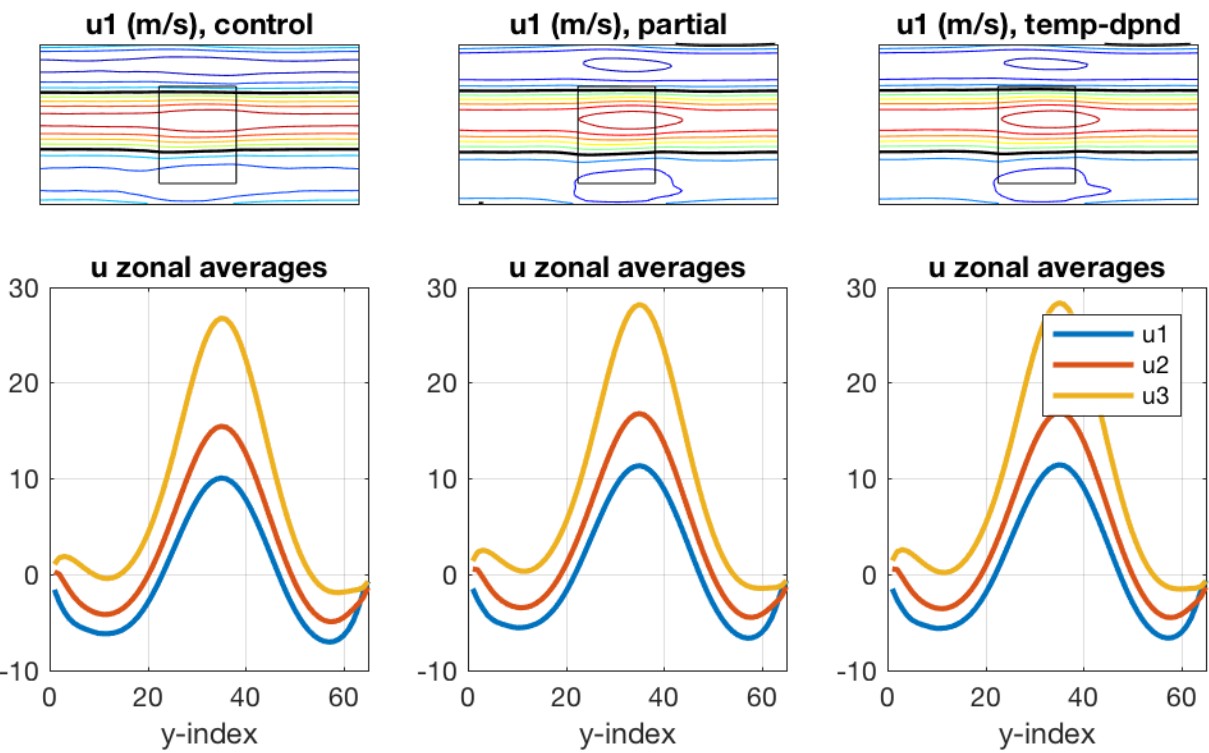

**Figure 1: Atmospheric mean state in control (left), partially coupled (middle) and fully coupled simulations (right) involving temperature dependent wind in the atmospheric mixed layer. Top: lower-layer zonal wind, CI=2 m s⁻¹, zero contour is black, rectangle in the middle (here and in other figures) marks the location of the ocean; bottom: zonally averaged zonal wind (m s⁻¹) in all layers (see legend).**






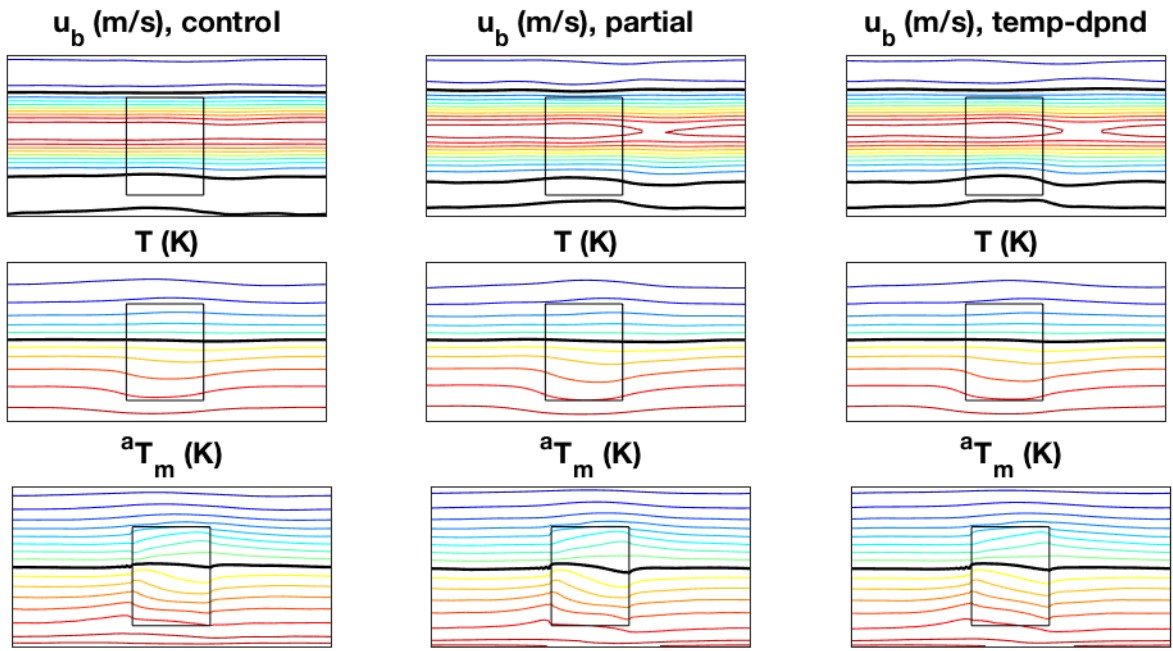

**Figure 2: Atmospheric mean state (continued). Top: barotropic zonal wind, CI=2 m s⁻¹; middle row: interior temperature**
**perturbation according to Eq. (12), CI=2 K; bottom: a.m.l. temperature, CI=2 K.**

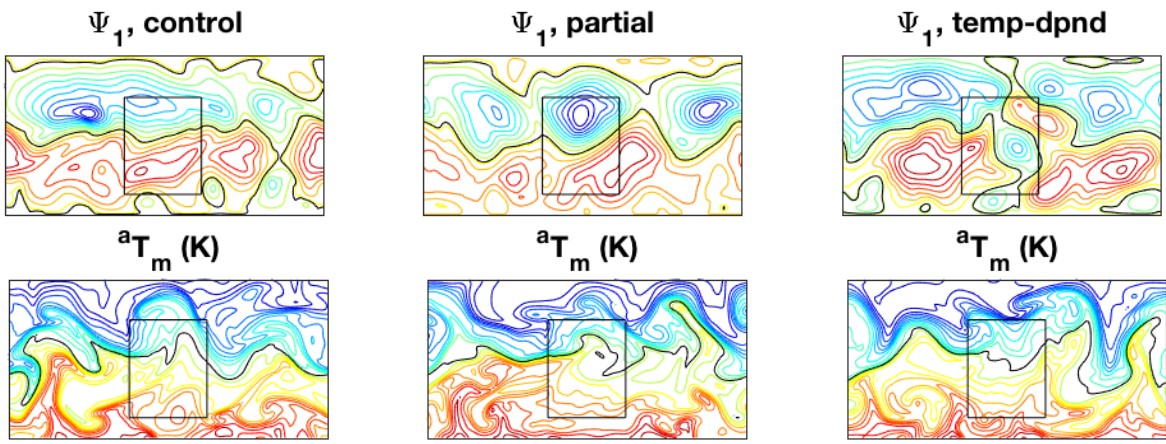

**Figure 3: Atmospheric snapshots from the three simulations. Top: lower-layer streamfunction; bottom row: a.m.l. temperature**
**perturbation, CI=2 K. Black curves show zero contour.**


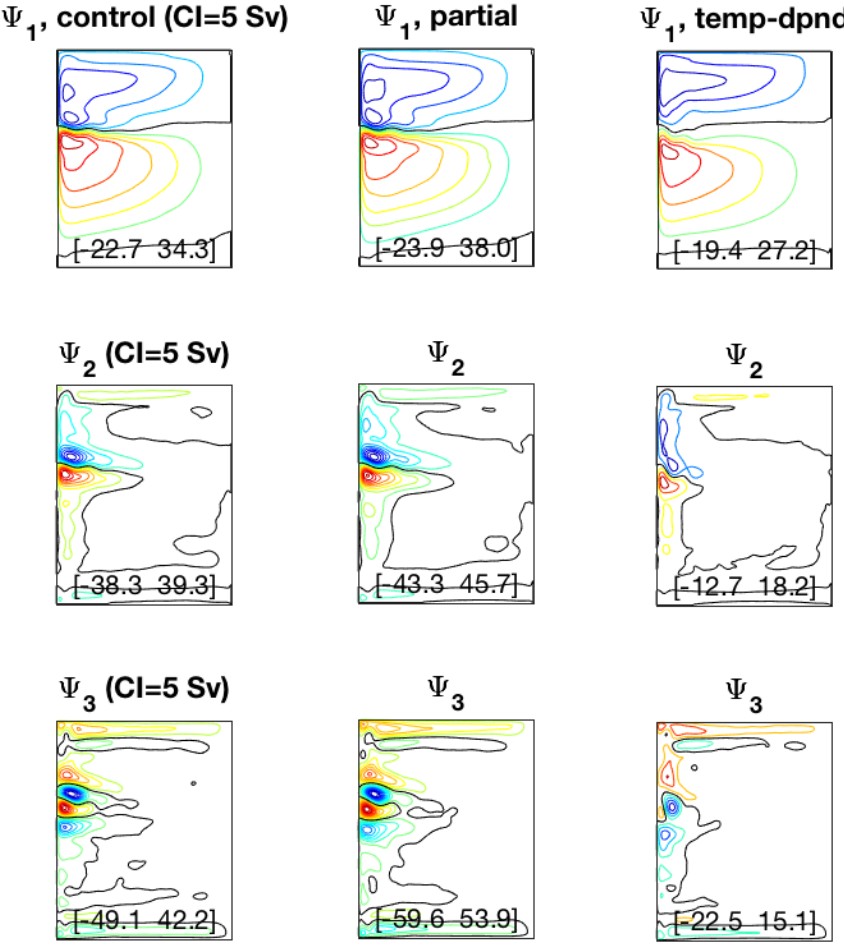

**Figure 4: Oceanic time-mean streamfunction (Sv) in control (left), partially coupled (middle) and fully coupled simulations (right) involving temperature dependent wind in the atmospheric mixed layer. Top, middle and bottom layer results are shown in the corresponding rows of the figure. CI is shown in panel captions. Black curves show zero contour. The panels also display the range**
**of streamfunction in Sv.**



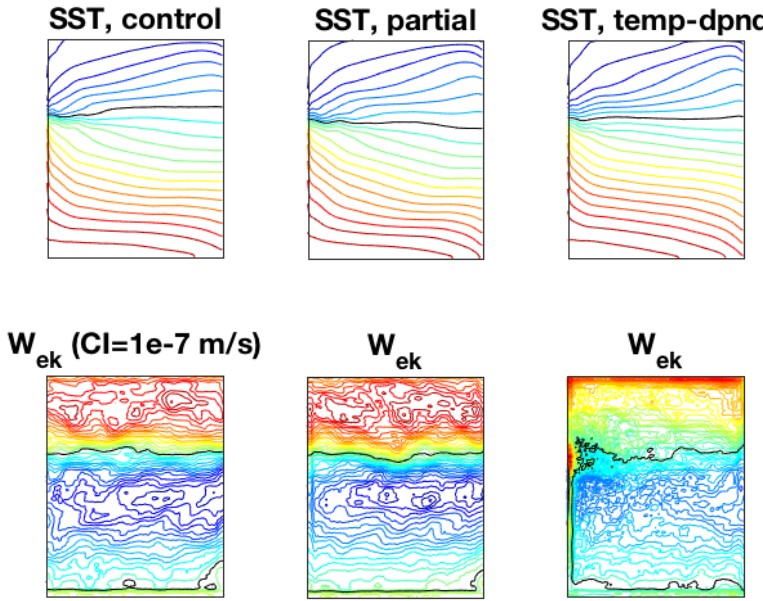

**Figure 5: Time-mean SST (top, CI=1K) and ocean Ekman pumping (bottom, CI=$10^{-7}$m s$^{-1}$) in control (left), partially coupled (middle) and fully coupled simulations (right) involving temperature dependent wind in the atmospheric mixed layer. Black line on SST plots shows –2°C anomaly with respect to the mean-state SST, approximately indicating the location of SST front; black line shows zero contour on Ekman pumping plots.**



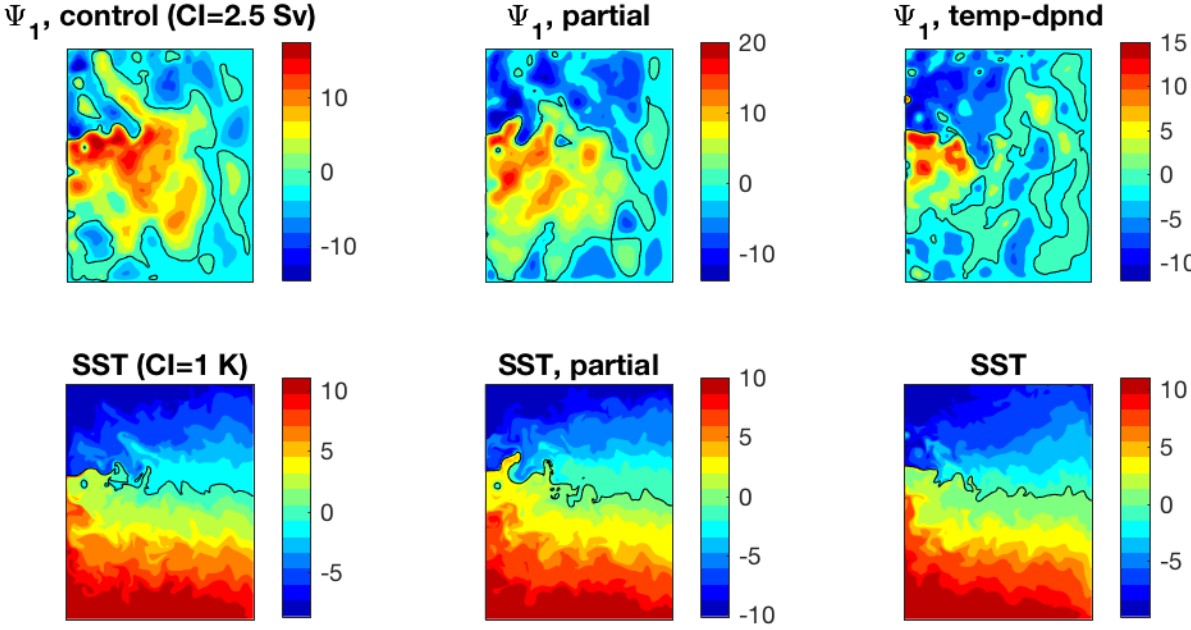

**Figure 6: Oceanic snapshots from the three simulations. Top: upper layer streamfunction (Sv); bottom: o.m.l. temperature perturbation (K).**

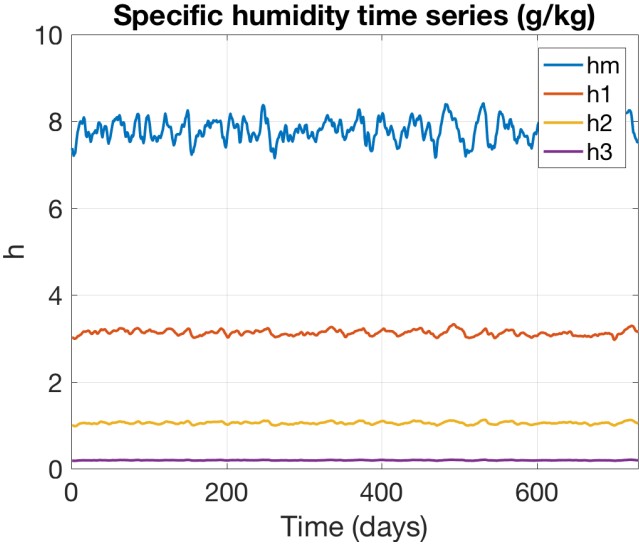

**Figure 7: Time series of the basin-mean specific humidity (g kg$^{-1}$) in the four atmospheric layers (see the legend).**





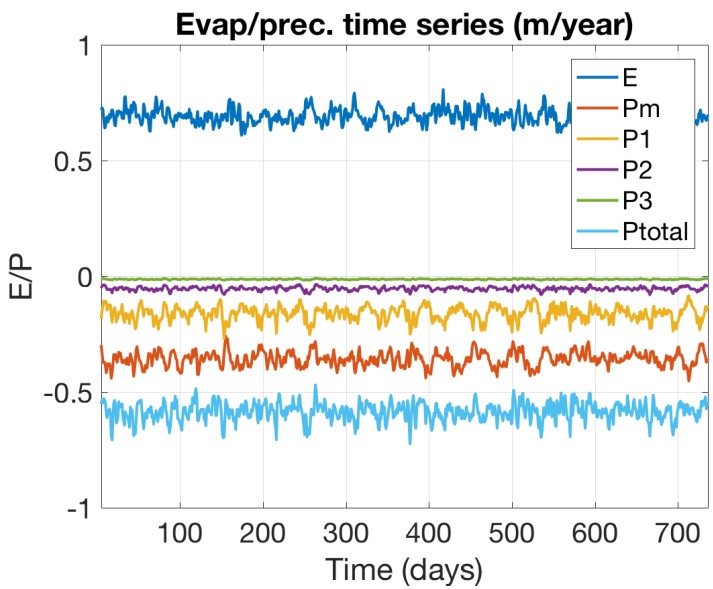


**Figure 8: Time series of the basin-mean evaporation (positive) and layer precipitation (negative) (m yr⁻¹); see the legend.**

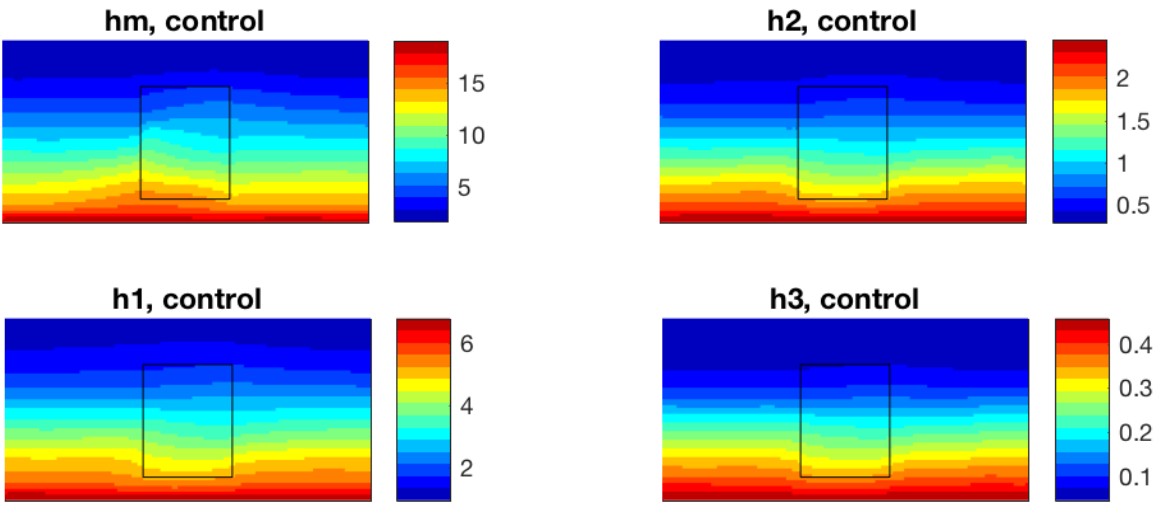

**Figure 9: Climatological distribution of specific humidity (g kg⁻¹).**






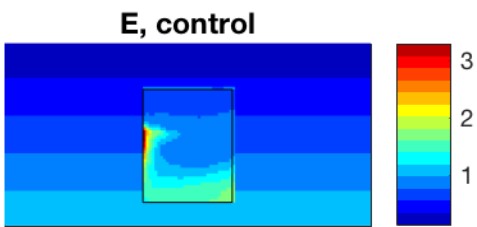

**Figure 10: Climatological distribution of evaporation (left) and precipitation in the mixed layer (right) (m yr⁻¹).**

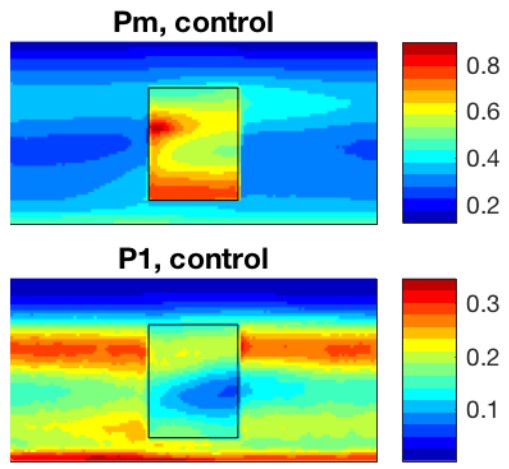

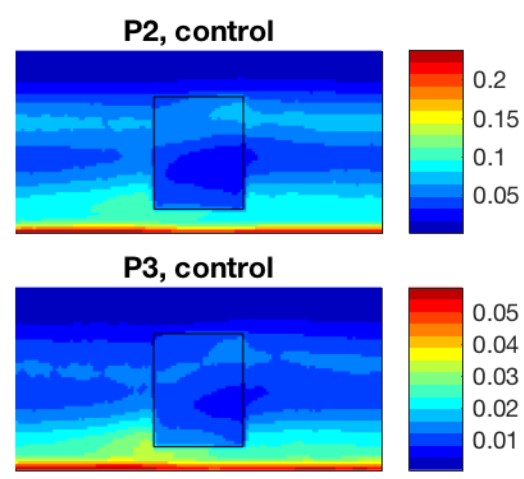

**Figure 11: Climatological distribution of precipitation (m yr⁻¹); all layers.**

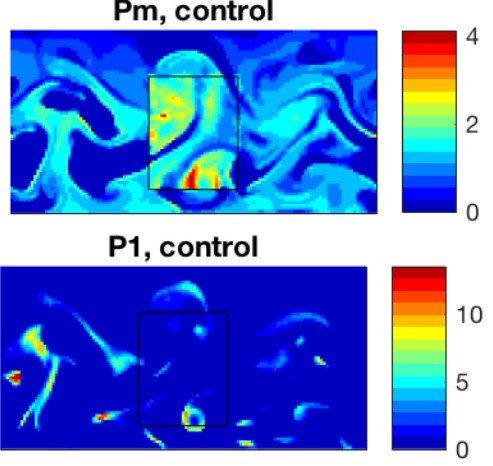

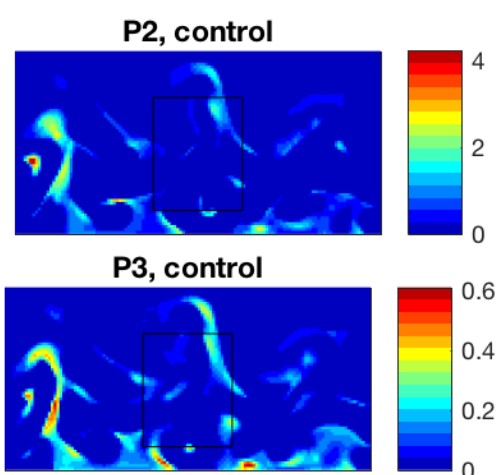

**Figure 12: Snapshot of precipitation (mm day⁻¹); all layers.**



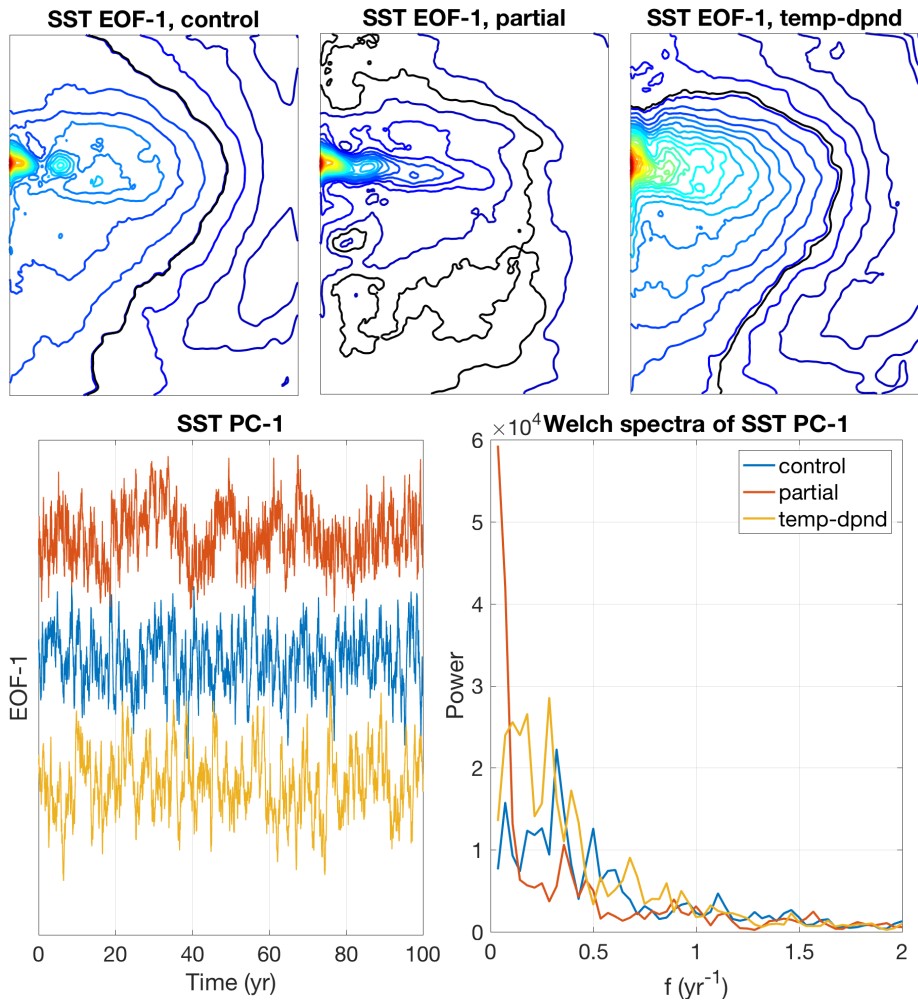

**Figure 13: The leading EOF of SST. Top row: EOF pattern in control (left), partially coupled (middle), and full temperature-dependent momentum coupling simulations (right); zero contour is black. Bottom row: PC-1 (left) and Welch-periodogram spectra (right) (the type of the simulation corresponding to each curve shown is given in the legend).**


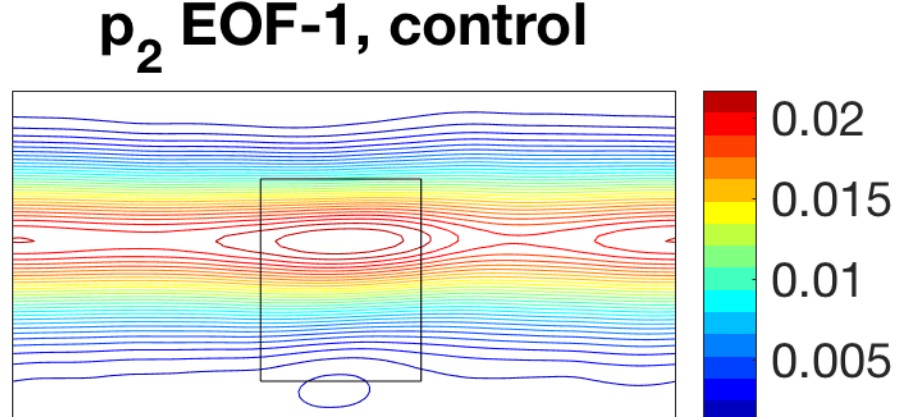

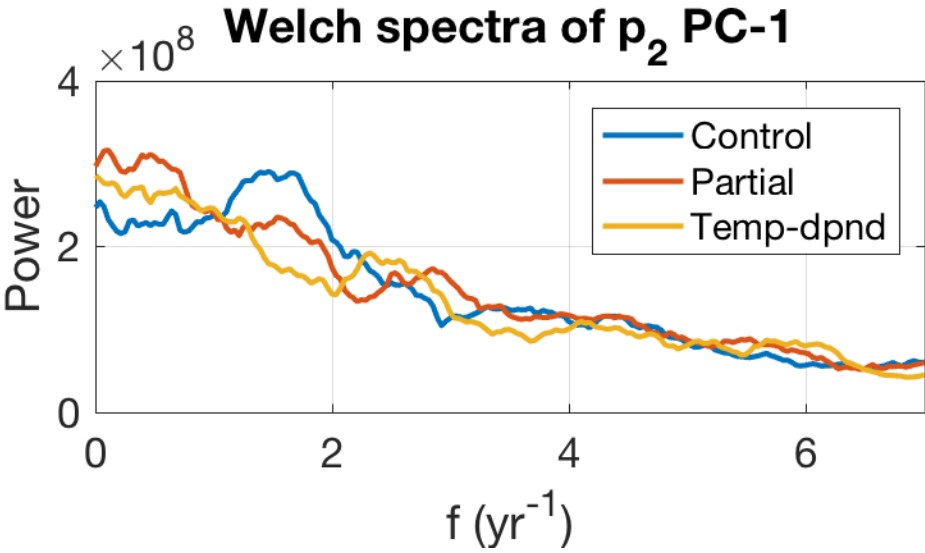

**Figure 14: The leading EOF of the mid-layer atmospheric streamfunction. Top: EOF pattern in the control simulation. Bottom: smoothed Welch-periodogram spectra of PC-1 in each simulation (the type of the simulation corresponding to each curve shown is given in the legend).**