# Peer review of "A Moist Quasi-Geostrophic Coupled Model: MQ-GCM2.0"

_Geoscientific Model Development, 2021_

## Author Response (AR1)

**Reply to reviewer comments on** "A Moist Quasi-Geostrophic Coupled Model: MQ-GCM2.0," by Kravtsov et al.

**Reviewer 1:**

This manuscript describes very significant upgrade of the Q-GCM model, which is unique and powerful modelling tool for various process studies in the midlatitude coupled ocean-atmosphere dynamics. The main new development is inclusion of moist dynamics, but there is a good list of other modifications. The paper is immaculately organized and written, and some interesting model simulations are included. The journal choice is also perfect. This is rare case when I suggest to accept this manuscript as it is. Being familiar with the previous model version and with many results obtained from its solutions, I am confident that the submitted work is of high quality and scientifically significant. I am looking forward to becoming one of the users of the new code (named MQ-GCM2) and to read future papers exploring various coupled flow regimes, as well as parameter and resolution dependencies.

We greatly appreciate positive comments of the reviewer and hope that this model will be useful to the community.

**Reviewer 2:**

I agree with the review comments from the first reviewer. As the first reviewer said, this paper is very well written and should be published in Geoscientific Model Development. This paper also shows significant improvement over the first version of Q-GCM model. Although I did not check the equations carefully, the authors included the moisture dynamics in this new version and made it a promising tool for studying air–sea interactions. The authors also gave a summary of the differences between the old and new version at the end of this paper. I would thank the authors for contributing this great work to the community. In my opinion, there are a few very minor concerns before publishing this manuscript:

We greatly appreciate positive comments of the reviewer and hope ourselves that this model will be useful to the community. Replies to the reviewer's detailed comments follow. We now rectify all of these points in the revised version of the paper (see below).

1. In Line 125, the authors said this model has n=3 layers in **oceanic and atmospheric modules**, but it looks to me that the atmospheric model does not only have 3 layers. Why do equations 1 and 2 only apply to the 1st and 3rd layer in the ocean model?

We placed a bar on top of the righthand side of k=1,3 to indicate that the equations apply for each value of k, from 1 to 3. The change was applied to equations (1), (2) [page 4] and (4), (5) [page 5].

2. In equations 18 and 31, the authors should explain the biharmonic term (Nabla^4). Is this term added to ensure the numerical stability or to resolve the physics?

The term is mainly included for numerical stability. The sentence to this effect was added after equations (18) [p. 11] and (31) [p. 16].

3. Line 440 is confusing to me. It seems that the authors run "control", "partially coupled" and "fully coupled" for both dry and moist models. But this paragraph is very confusing while I was reading. Why are you running three simulations for both dry and moist models? It also seems to me that the control run does not go for 130 years (as mentioned in the first sentence of this paragraph).

We rephrased this paragraph in the revised version of the paper to clarify these points (p. 19, first paragraph of section 5). We ran 6 simulations, each 130 years long (except for the control runs, which was, effectively, 230 years long each if one adds a 100-yr-long spin-up simulation from rest) and analyzed the final 100-yr time series of each. 3 of these simulations use the new version of the dry model, and the other 3 - the model with moist dynamics included. We run the simulations for each model to provide a preliminary assessment of the differences between the different versions of the model. Within the three simulations for either dry or moist version of the model, the first one - that we call "control" - is without SST feedback on AMBL wind speed; this is how the previous version of QGCM has been set up. The other two simulations include this feedback: "fully coupled" version includes full two-way feedback between the ocean and atmosphere, whereas in the "partially coupled" version the ocean only "sees" the Ekman pumping rates it would experience in the absence of the SST-dependent wind stress (but the APBL responds in the same way as in the fully coupled run) - please see eq. (26) and the associated discussion.

4. I would recommend the authors adding a few sentences introducing the validation/verification test in Section 5. Is it an idealized regional model or a realistic model? How is the boundary condition? How can this model validate the implementations in the model?

Section 5 documents key characteristics of the new model versions described in sections 1–4. Section 6 provides an outlook. In the revised version, we added an introductory sentence to section 5 to avoid confusion (l. 441).

---

## Author Response (AR2)

**Reply to editor's comments on** *"A Moist Quasi-Geostrophic Coupled Model: MQ-GCM2.0,"* by Kravtsov et al.

Dear author,

Thank you for the new version for the manuscript that globally addresses the remarks of the referees. But there are still the following issues I ask you to resolve before considering the paper for publication.
Thank you for considering those comments,
With best regards,
Sophie Valcke

Dear editor,

Thanks for your comments, which we addressed in the revised version of the paper. The specific replies follow.

Best regards,
Sergey Kravtsov and co-authors
* * *
Specific comments/Replies

First, your « data availability » section is still not right. In your reply to Juan A. Anel, you write: "MQ-GCM model is uploaded to Zenodo and DOI is https://doi.org/10.5281/zenodo.5250828. Also, the model's code is now under GNU General Public License v3.0 or later." But you did not change the « data availability » section accordingly in your manuscript. Please do so.

These lines have now been added at the start of the data availability section.

Then regarding the second remark of reviewer 2, I don't know if placing a bar on k=1,3 is a standard way to express that the equation applies for k=1 to k=3 ? Wouldn't k=[1,3] be clearer ? In all cases, you have to clarify this at line 133; writing explicitly "with k=1 to k=3" would remove any ambiguity. Please adapt the text.

We've changed the notations in both sets of equations to $k=1,2,3$ to avoid the potential confusion.

Finally,following a comment from the previous editor, shouldn't « involved » be changed for « evolved » at line 515 ?

We believe the current version with the term "involved" is correct (at least that's the meaning we intended), please see below the excerpt from a dictionary:

in·volved| inˈvälvd, inˈvôlvd | adjective — difficult to comprehend; complicated: *a long, involved conversation.*